# LEXam: Benchmarking Legal Reasoning on 340 Law Exams

**Yu Fan**[*E]   **Jingwei Ni**[*E,Z]   **Jakob Merane**[*E,L,M]   **Yang Tian**[*Z]
**Yoan Hermstrüwer**[Z,M]   **Yinya Huang**[E]   **Mubashara Akhtar**[E]   **Etienne Salimbeni**[O]
**Florian Geering**[Z]   **Oliver Dreyer**[S]   **Daniel Brunner**[C]   **Markus Leippold**[Z]
**Mrinmaya Sachan**[E]   **Alexander Stremitzer**[E]   **Christoph Engel**[M]
**Elliott Ash**[E]   **Joel Niklaus**[N]
[E]ETH Zurich   [Z]University of Zurich   [L]University of Lausanne
[M]Max Planck Institute for Research on Collective Goods   [O]Omnilex
[S]University of St. Gallen   [C]Swiss Federal Supreme Court   [N]Niklaus.ai
[*]Equal Contribution

## Abstract

Long-form legal reasoning remains a key challenge for large language models (LLMs) in spite of recent advances in test-time scaling. To address this, we introduce LEXam, a novel benchmark derived from 340 law exams spanning 116 law school courses across a range of subjects and degree levels. The dataset comprises 7,537 law exam questions in English and German. It includes both long-form, open-ended questions and multiple-choice questions with varying numbers of options. Besides reference answers, the open questions are also accompanied by explicit guidance outlining the expected legal reasoning approach such as issue spotting, rule recall, or rule application. Our evaluation on both open-ended and multiple-choice questions present significant challenges for current LLMs; in particular, they notably struggle with open questions that require structured, multi-step legal reasoning. Moreover, our results underscore the effectiveness of the dataset in differentiating between models with varying capabilities. Deploying an ensemble LLM-as-a-Judge paradigm with rigorous human expert validation, we demonstrate how model-generated reasoning steps can be evaluated consistently and accurately, closely aligning with human expert assessments. Our evaluation setup provides a scalable method to assess legal reasoning quality beyond simple accuracy metrics. Project page: `https://lexam-benchmark.github.io/`.

## 1 Introduction

Legal reasoning is a critical frontier for large language models (LLMs) specifically and artificial intelligence (AI) at large, requiring specialized domain knowledge and advanced reasoning abilities such as precedent interpretation, statutory analysis, and legal inference. Despite progress in general reasoning, legal reasoning remains difficult and under-assessed in NLP research. Moreover, the legal domain is inherently high-stakes and a failure to thoroughly examine the capabilities and limitations of models could lead to serious real-world consequences (Kleinberg et al., 2018; Medvedeva & Mcbride, 2023; Mahari et al., 2023). As LLMs are used increasingly for various legal tasks, such as legal judgment prediction (Wang et al., 2024a; Shui et al., 2023; Jiang & Yang, 2023; Deng et al., 2023), legal summarization (Bauer et al., 2023; Ash et al., 2024), legal interpretation (Savelka et al., 2023; Engel & McAdams, 2024; Dugac & Altwicker, 2025; Luo et al., 2025), legal case retrieval (Ma et al., 2024; Zhou et al., 2023; Mahari et al., 2024), legal argument classification (Thalken et al., 2023; Chlapanis et al., 2024), and contract drafting (Lam et al., 2023), it is imperative to assess and understand their capabilities and limitations.

However, LLM benchmarking on deterministic tasks has so far mainly concentrated in STEM domains, such as math Olympiad problems (Zheng et al., 2022; Liu et al., 2023; Tsoukalas et al., 2024; Ren et al., 2025) and physics problems (Feng et al., 2025; Zhang et al., 2025; Chow et al., 2025). Reasoning models such as OpenAI o3 (OpenAI, 2025) and DeepSeek-R1 (DeepSeek-AI, 2025a) have

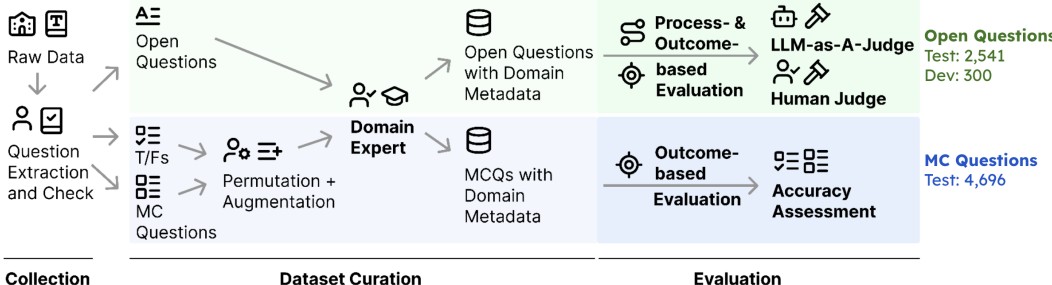

Figure 1: Process for creating LEXAM, a comprehensive legal reasoning benchmark derived from real law school exams. Created through careful expert extraction and curation, LEXAM contains 2,841 open-ended and 4,696 multiple-choice questions (MCQs), each with detailed domain metadata. Open-ended questions support both process- and outcome-based evaluation by LLMs-as-a-Judge and human judges, while MCQs provide clear, outcome-based assessments.

demonstrated impressive performance in tasks dominated by deduction and established scientific rules. Nevertheless, their abilities in more nuanced and context-dependent domains, such as the law, remain poorly understood (Engel et al., 2025; Niklaus et al., 2025). Legal reasoning requires not only rigorous deductive and inductive logic but also their application to complex practical scenarios, often involving ill-defined problems. This domain offers an ideal testbed for non-formal reasoning, yet it remains underexplored in the context of LLM performance. In contrast, STEM benchmarks allow for straightforward validation—for example, by checking final numerical answers or employing formal verifiers. Adopting this outcome-based evaluation paradigm, prior legal reasoning benchmarks (Guha et al., 2023; Fei et al., 2023; Chlapanis et al., 2024; Zheng et al., 2025) have primarily assessed the correctness of a model's final outputs, without explicitly evaluating the intermediate steps necessary for reaching a legally sound conclusion. Consequently, these benchmarks often overlook the reasoning process itself, leaving it unclear why LLMs fail to reach correct conclusions in some cases. This limitation raises concerns of tangible harm when LLM-based systems are deployed in practice.

Furthermore, while jurisdictions differ in their authoritative sources and treatment of precedent, they share a common foundation in legal reasoning, though with notable distinctions between common law and continental law systems. Common law systems (e.g. the United States, United Kingdom, and Australia) emphasize case precedent and judge-made law, whereas continental law systems (prevalent in most of Europe, Latin America, and parts of Asia) rely primarily on comprehensive legal codes and statutes. Across traditions, legal reasoning involves identifying the legal issue from the facts, determining the relevant rules, and applying those rules to the case. It also depends on diverse linguistic expressions that vary across systems. Prior studies (Ryan et al., 2024; Jin et al., 2025; Fan et al., 2025; Tian et al., 2026) show that LLMs exhibit cultural biases tied to language use. Evaluating LLMs on multilingual legal data is therefore essential to align them with real-world needs.

To address these gaps, we introduce LEXAM, a multilingual legal reasoning benchmark designed to assess the process-based and outcome-based correctness of the answers generated by LLMs, as illustrated in Figure 1. Based on exams in Swiss, European, and international law, LEXAM provides a comprehensive framework for evaluating the legal reasoning capabilities of LLMs through problem sets available in both English and German. The dataset comprises 7,537 law exam questions in English and German. It includes both long-form, open-ended questions and multiple-choice questions (MCQs) with varying numbers of options. Each open question is paired with reference answers and clear normative guidance, outlining the expected legal reasoning chain. This allows for thorough benchmarking of an LLM's legal reasoning capabilities beyond merely evaluating final answers. Questions are sourced from 340 law exams across 116 courses at the Bachelor's and Master's levels taught by leading law professors at the University of Zurich Faculty of Law, one of Europe's largest and most prestigious law schools (see Section B for further details). As a result, LEXAM spans a wide range of legal areas both in domestic and international law, thus providing a robust benchmark for the evaluation of legal reasoning skills (see Table 6 for more details).

Experimental results show that state-of-the-art (SOTA) models struggle with multi-step reasoning and applying legal rules and principles to novel scenarios. LLMs particularly struggle with open questions requiring structured legal reasoning. By designing and deploying an ensemble LLM-as-a-Judge evaluation framework, we demonstrate that model-generated reasoning steps can be evaluated

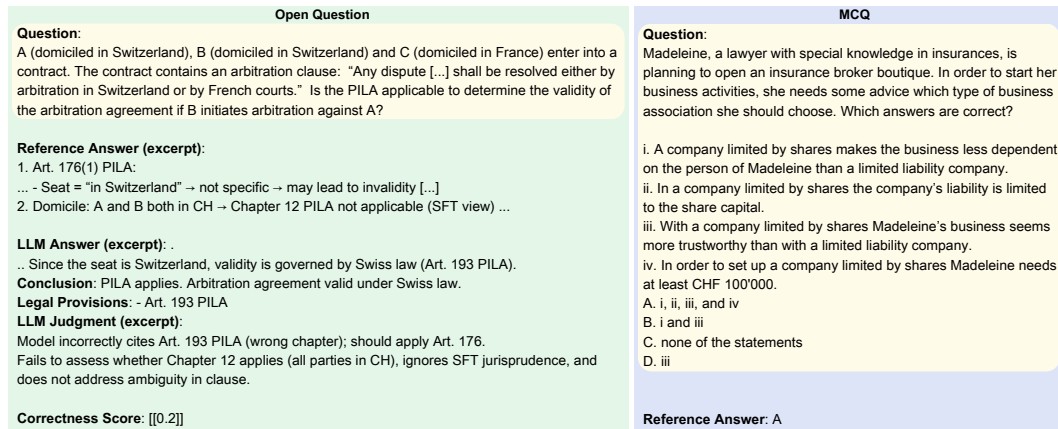

Figure 2: Illustration of a long-form open question (left, abbreviated example for illustration purposes. The full version is provided in Appendix E.3) and a MCQ-4 with a set of candidate statements (right).

consistently and accurately, closely aligning with human expert assessments. This approach provides a scalable method for assessing legal reasoning quality beyond simple accuracy metrics. Our work provides both a resource for NLP research in the legal domain and a practical tool for educators evaluating AI capabilities.

Our contributions can be summarized as follows: (1) We introduce a novel legal reasoning benchmark, addressing challenges such as long-context understanding, multilingualism, complex reasoning, and hallucinations. (2) We demonstrate baseline performance and challenges related to our benchmark by evaluating a wide range of SOTA models, providing a solid foundation for future research. (3) We perform expert analyses of model outputs, offering detailed insights into the dataset characteristics and current LLMs' limitations in legal reasoning. (4) Finally, we conduct a rigorous validation with human experts to establish a scalable and reliable LLM-based evaluation for open-ended questions. The resulting annotation can be reused to validate any backbone LLM for LEXAM evaluation.

## 2 DATASET

Figure 1 overviews our dataset construction process, detailed in the following sections.

### 2.1 CONSTRUCTION

The raw dataset is derived from 340 law exams that span 116 courses taught at the University of Zurich between 2016 and 2023 and was downloaded from their publicly accessible website. It contains a total of 3,628 questions, comprising 2,867 open questions, 398 MCQs, and 363 true/false questions (TFQs). All open questions, MCQs, and TFQs from the included courses were retained without filtering. Courses were included if they fell under one of four high-level legal areas, i.e., private law, public law, administrative law, or interdisciplinary, and were subsequently grouped by three legally trained authors into 78 subdomains based on doctrinal similarity, following the University's official course descriptions. The use and publication of the dataset was approved by the dean of the law school. Figure 2 provides a sample open question and a MCQ.

Although the exams originate from a Swiss institution, the dataset is not limited to Swiss civil-law doctrine. As detailed in the course metadata (Table 6), many of the courses cover international, comparative, and interdisciplinary topics that are central to both civil-law and common-law curricula. This diversity broadens the dataset beyond a single legal tradition and enables evaluation across multiple legal systems. Expanding the benchmark to include additional jurisdictions, in particular in common-law contexts, is an important future direction outlined in Appendix K.

Moreover, human performance data are unavailable due to institutional restrictions. In addition, the MCQ format does not appear in prior exams. We therefore report human performance in a separate experiment, summarized in Appendix J.

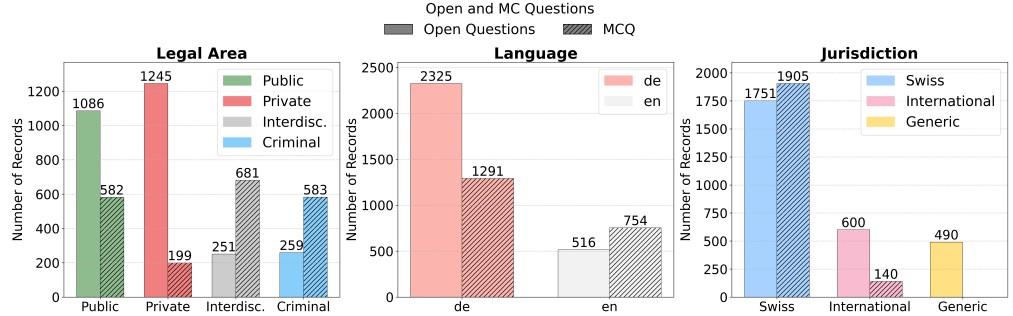

Figure 3: Distribution of open questions and MCQ-4 by legal area, language, and jurisdiction across development and test datasets. Open questions (solid) and MCQs (hatched).

### 2.1.1 OPEN QUESTIONS

We divided the open questions into test and development (dev) sets to support few-shot learning. Specifically, for each course containing at least ten open questions, we randomly selected five for the dev set and assigned the rest to the test set. This procedure yielded 300 questions from 60 courses in the dev set, and 2,541 in the test set.

### 2.1.2 MULTIPLE-CHOICE QUESTIONS

The MCQs were manually parsed into question stems and statements by three authors, where the stem provides contextual information to evaluate each statement. For each stem, we randomly generate MCQs that contain two to five statements extracted from the original MCQ. Each question includes one correct answer and 3 / 7 / 15 / 31 distractors, randomly selected from the set of all possible incorrect answers where applicable. The set of incorrect answers comprises all answer combinations where at least one statement is incorrect. This ensures a uniform number of answer choices per question and a baseline accuracy for a random guesser.

To unify the evaluation metric across formats, we convert all TFQs into multiple-choice format. First, each TFQ is manually parsed into a question stem (if applicable) and a statement. For questions that share the same stem, we group them and randomly generate MCQs with two to five associated statements. For stemless questions, we group them by domain and design templates to generate MCQs with two to five randomly selected statements. As with the original MCQs, each converted question includes four answer choices: one correct option and 3 / 7 / 15 / 31 randomly selected incorrect options from the entire pool of possible answers. This process yields a total of 4,696 MCQs. A sample MCQ-4 can be found in Figure 2.

While evaluating LLMs using MCQs has become common practice due to the ease of interpreting results, previous research has demonstrated that LLMs can exhibit unstable and sometimes unpredictable behavior in response to changes in question formatting, phrasing, or the order of answer choices (Wang et al., 2024b; Pezeshkpour & Hruschka, 2023; Alzahrani et al., 2024). To further examine this issue, we constructed an auxiliary MCQ dataset comprising 385 questions, each containing five statements. This subset is determined solely by the structural requirement of having five statements, with no additional filtering applied, and the questions originate from ten courses (see Table 8 for the distribution). We systematically vary the number of answer choices (4, 8, 16, and 32) while keeping both the question stem and the order of the statements constant. This perturbation setup aims to diagnose whether models select the correct answers based on genuine domain understanding or merely guess due to insufficiently challenging distractors.

### 2.2 DATASET STATISTICS AND STRUCTURED FEATURES

The LEXAM dataset comprises 2,841 open questions, split into dev (10.6%) and test (89.4%) sets, and 4,696 MCQs with vary numbers of options (4 to 32). Detailed per-course statistics and textual descriptions are in Section B.2. Figure 3 illustrates the data distributions.

We stored several features as metadata for our analysis. The `area` feature groups questions into four distinct legal categories: private law, public law, criminal law, and interdisciplinary. The `course`

feature represents course information, while the `jurisdiction` feature classifies questions into one of three legal jurisdictions: Swiss, international, or generic. In addition, the `language` feature indicates the language of the exams, either English or German. The English and German questions are independent items drawn from distinct exams and are not parallel translations. We deliberately avoided translation due to the complexity and risk of semantic drift in legal contexts (see also (Niklaus et al., 2025)). The binary label `none_as_an_option` identifies MCQs where the option "None of the statements" ("Keine der Aussagen") appears; this attribute is applicable exclusively to MCQs. Another feature, `n_statements`, specifies numerically how many statements each MCQ includes, again relevant only for MCQs. furthermore, the binary label `negative_question` indicates whether the question is phrased negatively (e.g. "Which of the following statements are incorrect?"), which is once again relevant only for MCQs. Lastly, the feature `year` specifies when the exam was administered.

## 3 EXPERIMENTAL SETUP

In this section, we cover the experimental details used to benchmark models on LEXAM questions.

### 3.1 OPEN QUESTIONS

Section E.1 exhibits the prompt to answer open questions. It first informs about the relevant knowledge domain with course name, and then provides standard guidance that law students follow when answering exam questions.

**Evaluation Metric.** Evaluating open questions in specialized domains like law is challenging. Lexical or shallow semantic metrics, such as BLEU (Papineni et al., 2002), ROUGE (Lin, 2004), BERTScore (Zhang et al., 2020), and AlginScore (Zha et al., 2023), may not accurately reflect answer quality, as high lexical or semantic overlap does not necessarily indicate correct legal reasoning, and vice versa. Legal reasoning often involves low lexical similarity (Zheng et al., 2025). LLM-as-a-Judge (Zheng et al., 2023) may help, but whether a general-domain LLM is capable of judging the quality of legal responses remains an open question.

To develop a more suitable LLM judge and verify its reliability, we proceed as follows: First, two of the authors, both with formal doctoral-level legal training, draft a specialized judging prompt, as presented in Section E.2. Then, they conduct a pilot study to iteratively optimize the prompt (with GPT-4o as the judging LLM) until they were satisfied with the judging performance. This pilot was conducted on a diverse sample of courses and a key challenge was to calibrate penalties. Through this process, several refinements were also made to the prompt.Rather than encoding an abstract theory of legal reasoning, the prompt is tuned to track adherence to the doctrinal structure present in professor-written model answers and to penalize domain-specific errors such as fabricated or incorrect statutory citations. In this way, legal expertise enters the system both through these model answers and through the domain-specific prompt design, which distinguishes our setup from general-purpose reasoning evaluation.

The next step is to identify capable LLMs for executing the expert-verified prompt. We rigorously evaluate various LLMs against human experts using the Alternative Annotator Test (Alt-test) (Calderon et al., 2025), detailed in Section 5. We find that only proprietary LLMs (e.g., GPT-4o, and Gemini-2.5-Pro) and extremely large reasoning models (e.g., DeepSeek-R1) consistently outperform human judges. Exclusive reliance on such models for evaluation, however, undermines the accessibility of LEXAM. Encouragingly, we find that carefully constructed ensembles of open-source LLMs—for instance, taking the minimum score of Qwen3-32B and DeepSeek-V3—also surpass human judges. Our analyses further demonstrate that minimum-score ensembles of LLM judges consistently enhance evaluation performance.

In our evaluation, we adopt an minimum-score ensemble of GPT-4o, Qwen3-32B, and DeepSeek-V3 to grade open questions. Practitioners may instead rely on any LLM judge or ensemble that passes the Alt-test.

Importantly, our ensemble design avoids dependence on any single model, whether open or closed. The adopted judge includes two open-source models (Qwen3-32B and DeepSeek-V3) and one closed model (GPT-4o), ensuring robustness across different architectures and providers. This

diversity is crucial for reducing self-bias and family-bias, which are known risks in LLM-as-a-Judge setups. By aggregating pointwise minimum scores, our ensemble suppresses overly favorable evaluations for answers produced by the same model or model family (see Appendix G.6). We also report performance for fully open-source judge ensembles in Appendix G.4, enabling reproducible evaluation pipelines without reliance on proprietary systems. Additional robustness checks for LLM judgese are presented in Appendices H and I. While we do not present the ensemble LLM-as-a-Judge approach as a novel evaluation paradigm, we argue that it provides a practical, empirically validated solution for reliable evaluation of open-ended legal reasoning. The ensemble is designed specifically to mitigate concerns around self-referentiality and bias, as supported by both human comparisons (Alt-test) and consistency checks.

## 3.2 MULTIPLE-CHOICE QUESTIONS

Appendix E.4 shows the prompt used for answering MCQs. It first provides the course title to indicate the relevant domain, then guides the LLM through standard legal reasoning steps such as clarifying facts, identifying issues, and explaining rules.

**Evaluation Metric.** We use accuracy scores for the MCQ evaluations. Since the choice label distribution is balanced through permutation, accuracy reflects unbiased performance of LLMs. For example, a random predictor will achieve approximately 25% accuracy in our MCQ dataset with four answer choices. To assess the stability of the aggregated accuracy, we also compute its standard error by bootstrapping across different sampling of subsets. As a robustness check, we also evaluate a subset of frontier models on perturbed MCQs (see Section 2.1.2).

## 3.3 LLM SETTINGS

**Model Selection.** For open questions, we evaluate 35 LLMs across three representative categories: (1) *Reasoning* LLMs include DeepSeek-V3.2-reasoner (Liu et al., 2025), DeepSeek-V3.2-Exp (DeepSeek-AI, 2025), DeepSeek-R1 (DeepSeek-AI, 2025a), GPT-5, -mini, and -nano (OpenAI, 2025b), Gemini-2.5-Pro (Gemini, 2025), Gemini-3.0-Pro-preview[1], Claude-3.7- (Anthropic, 2025) and 4.5-Sonnet (Anthropic, 2025), GPT-OSS-20B and -120B (Agarwal et al., 2025), O3-mini (OpenAI, 2025d), QwQ-32B (Qwen, 2024), Qwen3 32B and 235B (Qwen, 2025), and Qwen 3 Next (AI, 2025a); (2) *Large* models include GPT-4.1 (OpenAI, 2025a), GPT-4o (OpenAI, 2024a), DeepSeek-V3.2-chat (Liu et al., 2025), DeepSeek-V3 (DeepSeek-AI, 2025b), Llama-4-Maverick (Meta AI, 2025), and Llama-3-Instruct(it) with 70B and 405B parameters (Grattafiori et al., 2024); and (3) *Small* models include GPT-4.1-mini, and -nano (OpenAI, 2025a), GPT-4o-mini (OpenAI, 2024b), Gemma-3-12B-it (Gemma, 2025), Gemma-2-9B-it (Gemma, 2024), Phi-4 (Abdin et al., 2024), EuroLLM-9B-it (Martins et al., 2024), Qwen-2.5-7B-it (Qwen, 2025), Ministral-8B-it (Mistral AI, 2024), Llama-3.1-8B-it (Meta-llama, 2024), and Apertus-70B and -8B (Apertus et al., 2025). We select well-performing flagship LLMs from diverse families (e.g., Llama, GPT) and also prioritize those with strong multilingual abilities (e.g., EuroLLM, Gemma-3) due to LEXAM's multilingual nature. For MCQ-4 we selected a subset of the aforementioned models; for MCQ-16, we selected a subset of the aforementioned models and several newer models, including GPT5.2 (OpenAI, 2025c), Claude-4.6-Sonnet (Anthropic, 2026), Qwen3.5-plus (AI, 2025b), GLM-5 (Z.AI, 2026), Kimi-k2.5 (Moonshot AI, 2026), Gemma-3-27B, and MiniMax-m2.5 (MiniMaxAI, 2026).

**Inference Hyperparameters.** For conventional LLMs, we set temperature to 0 and max length to 4096, which is sufficient for complete answers. We adopt the lighteval framework (Fourrier et al., 2023) to standardize the evaluation of conventional LLMs with different endpoints. Reasoning models have more diversified sets of hyperparameters and are not supported by lighteval at the time of writing, so we follow the official recommended settings. We document detailed settings in Section F.

## 4 RESULTS

### 4.1 RESULTS ON OPEN QUESTIONS

---

[1]https://deepmind.google/models/gemini/pro/

We show our main results on open questions in Table 1. [2] SOTA reasoning models, such as GPT-5 and Gemini-2.5-Pro, achieve the best performance, with GPT-5 showing the highest average score (70.20). Other reasoning models, namely Claude-3.7-Sonnet, Claude-4.5-Sonnet, GPT-5-mini, and DeepSeek reasoning models, are also competitive among all the models, highlighting the strength of models explicitly optimized for reasoning tasks. Among the large conventional LLMs, GPT-4.1 and GPT-4o perform strongly, with 57.50 and 56.93, respectively, followed by DeepSeek models and Llama-4-Maverick. Small LLMs generally exhibit the lowest overall performance. However, GPT-4.1-mini stands out with an average score of 54.58. Interestingly, Gemma-3-12-it achieves a score of 41.29, comparable to Llama-3.1-405B-it (43.14) and Llama-3.3-70B-it (41.27), despite being approximately $33\times$ and $6\times$ smaller in size, respectively. This strong performance may be attributable to the model's particular focus on multilinguality. Moreover, smaller or older-generation LLMs, including reasoning models such as QwQ-32B and O3-mini, as well as conventional models like GPT-4o-mini, Llama-3.1-405B-it, and Llama-3.3-70B-it, generally perform worse than flagship LLMs. Notably, bootstrapped standard errors remain low across all groups, suggesting that the LLMs perform consistently across different subsets of the dataset. We provide a more in-depth qualitative analysis of common error patterns in Appendix I.

Figure 4 shows average LLM-Judge scores across three dimensions: *Language*, *Legal Area*, and *Jurisdiction*. Additional features appear in Appendix C.1 (Figure 9). Reasoning models consistently outperform large and small models universally. For *Language*, all groups perform better on English than German courses, with small models showing the largest gap. The English-German performance gap likely reflects both linguistic and legal reasoning differences. Since the questions are not parallel translations, disentangling these factors would require high-quality legal translation, which is an important but out-of-scope direction we discuss in Appendix K. In *Legal Area*, interdisciplinary and public law topics score higher than criminal and private law. For *Jurisdiction*, generic and international law tasks achieve higher accuracy than Switzerland-specific ones.

Table 1: Performance of models on long-form open questions with bootstrapped standard error. All questions are graded by ensemble LLM judges (GPT-4o, DeepSeek-V3, & Qwen3-32B). Results are sorted by Judge Score in descending order.

| | Model | Open Questions Judge Score (± S.E) |
|---|---|---|
| Reasoning | GPT-5 | **70.20 (± 0.41)** |
| | Gemini-2.5-Pro | 67.40 (± 0.51) |
| | Claude-3.7-Sonnet | 62.86 (± 0.51) |
| | Claude-4.5-Sonnet | 62.76 (± 0.43) |
| | GPT-5-mini | 60.32 (± 0.45) |
| | DeepSeek-V3.2-Exp | 57.42 (± 0.45) |
| | DeepSeek-V3.2-reasoner | 56.53 (± 0.45) |
| | DeepSeek-R1 | 55.91 (± 0.51) |
| | Gemini-3-Pro-preview | 55.38 (± 0.64) |
| | GPT-OSS-120B | 51.74 (± 0.46) |
| | O3-mini | 48.13 (± 0.49) |
| | Qwen3-235B | 47.25 (± 0.46) |
| | QwQ-32B | 44.36 (± 0.53) |
| | Qwen3-Next | 43.37 (± 0.48) |
| | Qwen3-32B | 40.00 (± 0.43) |
| | GPT-OSS-20B | 32.12 (± 0.37) |
| | GPT-5-nano | 27.25 (± 0.63) |
| Large | GPT-4.1 | **57.50 (± 0.51)** |
| | GPT-4o | 56.93 (± 0.48) |
| | DeepSeek-V3.2-chat | 55.99 (± 0.45) |
| | DeepSeek-V3 | 52.53 (± 0.48) |
| | Llama-4-Maverick | 47.25 (± 0.46) |
| | Llama-3.1-405B-it | 43.14 (± 0.41) |
| | Llama-3.3-70B-it | 41.27 (± 0.41) |
| | Apertus-70B | 34.70 (± 0.39) |
| Small | GPT-4.1-mini | **54.58 (± 0.43)** |
| | GPT-4.1-nano | 43.68 (± 0.41) |
| | GPT-4o-mini | 42.55 (± 0.39) |
| | Gemma-3-12B-it | 41.29 (± 0.48) |
| | Phi-4 | 38.54 (± 0.42) |
| | Gemma-2-9B-it | 27.41 (± 0.37) |
| | EuroLLM-9B-it | 22.95 (± 0.35) |
| | Apertus-8B | 22.44 (± 0.41) |
| | Qwen2.5-7B-it | 16.67 (± 0.29) |
| | Ministral-8B-it | 14.88 (± 0.32) |
| | Llama-3.1-8B-it | 10.00 (± 0.26) |

## 4.2 RESULTS ON MULTIPLE-CHOICE QUESTIONS (16 CHOICES)

As shown in Table 9, reasoning models lead in accuracy, with GPT-5.2 (52.53%) and Claude-4.6-Sonnet (52.42%) achieving the top two scores. These are closely followed by Qwen3.5-plus (43.77%) and DeepSeek V3.2-reasoning (41.15%), which also demonstrates strong competitive performance. Among large and small models, Llama-3.3-70B-it (26.07%) and Phi-4 (21.40%) outperform the rest, while all others in the two categories score below 20%. Across all groups, bootstrap standard errors are low (typically below 1.5%), indicating stable performance estimates.

Figure 5 shows model accuracy across five dimensions: *Language*, *Legal Area*, *Jurisdiction*, *Number of Statements* (legal propositions preceding multiple choice options), and *Question Polarity* (affirmative vs. negative phrasing). Further results on MCQ-4 appear in Appendix C.2 Figure 10. Reasoning models consistently outperform others, with the largest gap in English-language courses

---

[2] We will present results from future models at `https://lexam-benchmark.github.io/#leaderboard`.

(A), international jurisdictions (C), and affirmative questions (E). All models show accuracy drops for German courses, Swiss law and negative questions, while public law yields lower accuracy than private or interdisciplinary law (B). When the number of statements increases from 4 to 5 (D), reasoning models maintain comparatively stable performance while large and small models degrade, indicating sensitivity to prompt length. Interestingly, all models exhibit a significant performance decrease when the MCQ is phrased negatively. This effect is particularly pronounced for reasoning models, which is counterintuitive. The performance of small models on negatively phrased questions is approximately random. All bars indicate bootstrapped standard errors.

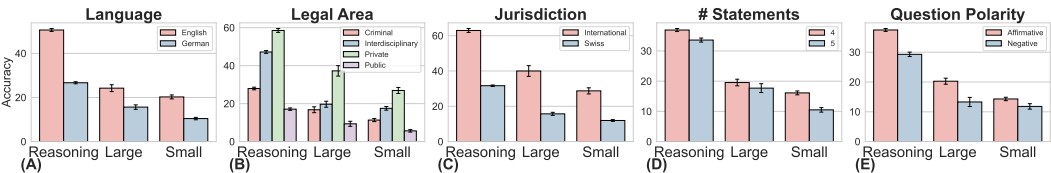

Figure 5: Model performance on MCQ-16 grouped by various metadata.

### 4.2.1 ROBUSTNESS CHECK ON MULTIPLE-CHOICE QUESTIONS WITH PERTURBATIONS

The experiments on MCQs with perturbations were conducted using seven selected models: DeepSeek-R1, DeepSeek-v3, Gemini-2.5-Pro, Claude-3.7-Sonnet, o3-mini, GPT-4.1, and GPT-4o. The models were tested using the same combinations of question stems and statements, with a varying number of choices: 4, 8, 16, and 32. The accuracy results, summarized in Table 2 and illustrated in Figure 11, clearly indicate a consistent trend

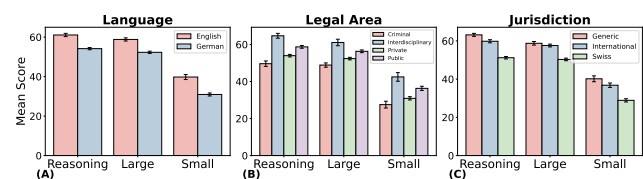

Figure 4: Model performance on open questions grouped by metadata.

across all settings. Specifically, the accuracy decreases significantly as the number of answer choices increases.

In all settings tested, the models show significant performance differences, further confirming the dataset's effectiveness in distinguishing between models with different capabilities. Further details are summarized in Appendix D. These results indicate substantial variability in robustness and accuracy among the evaluated models, highlighting risks associated with performance sensitivity stemming from reliance on spurious signals and guessing behavior. Moreover, these findings urge caution when benchmarking LLMs using standard MCQs, as this approach may produce overly optimistic results.

## 5 EXPERT EVALUATION FOR OPEN QUESTIONS

Evaluating open-ended legal questions presents a complex challenge. Prior work has increasingly turned to LLM-based evaluators ("LLM-as-a-judge"), yet concerns remain about their reliability in high-stakes domains such as law. To address this, we introduce an ensemble LLM judge (see Section 3.1

Table 2: Model Accuracy (Acc.) and Bootstrap Standard Error (S.E.) across varying context lengths as percentages.

| Model | 4 Choices | 8 Choices | 16 Choices | 32 Choices |
|---|---|---|---|---|
| Gemini-2.5-Pro | **68.61 (± 2.37)** | **51.56 (± 2.55)** | **45.24 (± 2.43)** | **35.62 (± 2.30)** |
| Claude-3.7-Sonnet | 60.92 (± 2.48) | 48.59 (± 2.44) | 40.38 (± 2.43) | 33.02 (± 2.31) |
| DeepSeek-R1 | 57.54 (± 2.49) | 44.11 (± 2.45) | 36.94 (± 2.36) | 24.93 (± 2.21) |
| GPT-4.1 | 58.02 (± 2.53) | 42.31 (± 2.51) | 33.23 (± 2.38) | 26.30 (± 2.13) |
| GPT-4o | 53.73 (± 2.56) | 36.42 (± 2.37) | 22.55 (± 2.09) | 21.81 (± 2.18) |
| DeepSeek-V3 | 58.57 (± 2.57) | 36.07 (± 2.40) | 28.92 (± 2.28) | 16.03 (± 1.88) |
| o3-mini | 50.02 (± 2.55) | 33.54 (± 2.34) | 24.46 (± 2.16) | 17.01 (± 1.89) |

and Appendix G.4 for more details) based on GPT-4o, Qwen3-32B, and DeepSeek-V3. To rigorously validate its use, we conduct an expert evaluation. This analysis both benchmarks human annotator agreement and assesses whether our LLM judge can serve as a statistically grounded proxy for expert annotation using the Alt-test (Calderon et al., 2025).

## 5.1 INTER-ANNOTATOR AGREEMENT

We first measured the consistency of three legal experts in rating 50 randomly drawn question–answer pairs on a 0-10 scale. All three hold a Swiss Master's degree in law and are pursuing or have completed a doctorate in Law. The experts scored all 50 items blindly and independently, following only brief calibration instructions (see Appendix G.1 for details).

The three legal experts achieve an average Pearson correlation of $r = 0.70$, indicating strong linear consistency in their ratings. The mean quadratic weighted $\kappa = 0.49$ further confirms substantial agreement. A mean absolute error of 1.95 points on the 0–10 scale implies that expert judgments deviate by less than two points on average. For the full pairwise breakdown, see Appendix G.3.

## 5.2 ALTERNATIVE ANNOTATOR TEST

To rigorously assess whether our ensemble LLM judge can replace human experts, we applied the Alt-test (Calderon et al., 2025) with $\varepsilon=0$ (i.e. no bonus or tolerance margin for LLM judges). Following their guidelines, we used a leave-one-out design with $m = 3$ experts and $n=50$ items. We reused the exact same 50-item random sample and three expert annotators described in Section 5.1.

In each of three leave-one-out trials, we compared our ensemble LLM judge's distance to the mean of the other two experts against that expert's distance, counted "wins" for the LLM versus the human, formed a difference vector $d_i \in \{-1, +1\}$, and ran a one-sided t-test on $\bar{d} < \varepsilon$. We then corrected the three raw $p$-values via Benjamini–Yekutieli (BY) at $\alpha=0.05$.

Our ensemble LLM judge surpasses the recommended threshold of $\omega \geq 0.5$. As Table 3 shows, it significantly exceeds the three experts, yielding a winning rate $\omega=1.00$. Additional Alt-test results using other LLMs are provided in Section G.4.

Table 3: Results of the Alternative Annotator Test with $\varepsilon = 0$.

|  | LLM wins | Human wins | raw $p$ | BY $p$ | Adv. Prob. $\rho$ |
|---|---|---|---|---|---|
| Legal Expert 1 | 36 | 14 | 0.001 | 0.002 | 0.72 |
| Legal Expert 2 | 34 | 16 | 0.005 | 0.009 | 0.68 |
| Legal Expert 3 | 37 | 13 | 0.000 | 0.001 | 0.74 |
| **Winning Rate $\omega$** | | | | **1.00** | |

Taken together, our expert evaluation establishes that annotators agree substantially among themselves (Pearson $r=0.70$, quadratic weighted $\kappa=0.49$, $MAE=1.95$) and that our ensemble LLM judge matches ($\omega=1.00$) three annotators even under the strictest no-bonus criterion ($\varepsilon=0$). These results validate our ensemble LLM judge as a statistically justified replacement for open questions.

## 5.3 FURTHER ROBUSTNESS ANALYSES OF LLM JUDGES

We conducted additional quantitative and qualitative analyses to assess the robustness of judge models across languages and legal domains using four models (GPT-4o, Gemini-2.5-Pro, DeepSeek-R1, and Claude-4-Sonnet). Correlation tests on expert-annotated samples show that all models align closely with human scores in both English and German, with only minor and statistically insignificant language effects, indicating cross-linguistic consistency (see Tables 15 and 16). Qualitative inspection of LLM answers revealed recurring failure modes, including flawed legal reasoning, references to incorrect or fictitious authorities, and weaker German or multilingual proficiency in smaller models, which often resulted in incoherent outputs. We also observed systematic differences in leniency: GPT-4o was the most generous, Claude-4-Sonnet and Gemini-2.5-Pro were stricter, and DeepSeek-R1 fell in between. Notably, the largest gaps between model and human evaluations appeared in cases with higher human disagreement, underscoring the inherent difficulty of borderline legal judgments. These results further motivate the use of an ensemble judge with minimum-score integration. More details are available in Appendices H and I.

# 6 RELATED WORK

## 6.1 INTERNATIONAL BENCHMARKS

Multiple international benchmarks for legal reasoning have been established, comprising various individual datasets, primarily sourced from China and the United States. Examples are LegalBench

Table 4: Comparison of LEXAM to prior legal QA datasets. Answer type is either true-false (TF), multiple-choice (MC), or long-form (LF), where we specify the average answer length in whitespace-split words in brackets. CH is short for Switzerland. Int. is short for International.

| Benchmark | License | Jurisdiction | Languages | Legal Domain | # Examples | Answer Type |
|---|---|---|---|---|---|---|
| Housing Statutes | Unknown | US | English | Statutory Housing Law | 6,853 | TF |
| Sara | Unknown | US | English | Tax Liability | 160 | TF |
| Brazilian Bar Exams | Unknown | Brazil | Portuguese | 17 law areas | 2,130 | MC |
| COLIEE | CC BY 4.0 | Japan, Canada | Japanese, English | Japanese Civil Law, Canadian Case Law | 1,774 | MC |
| GLOBALCIT | CC BY 4.0 | International | English | International Citizenship Law | 9,310 | MC |
| JecQA | CC BY-NC-ND | China | Chinese | Chinese Law | 21,072 | MC |
| MMLU (Legal Subset) | CC BY 4.0 | Primarily US | English | General Legal Knowledge | 1,970 | MC |
| Multistate Bar Exam | Unknown | US | English | Bar Exam questions | 1,195 | MC |
| PrivacyQA | MIT | Global | English | Privacy Law & Data Protection | 1,750 | LF (140) |
| LEXAM (ours) | CC BY 4.0 | CH, EU & Int. | German, English | 78 Subdomains | 4,886 | LF (248) & MC |

(Guha et al., 2023), LawBench (Fei et al., 2023), LexEval (Li et al., 2024), LEXTREME (Niklaus et al., 2023b) or LexGLUE (Chalkidis et al., 2022). While they propel the field forward, they still fall short in the complexity of the tasks evaluated. In contrast, we introduce very challenging law exam questions, often requiring detailed, long-form answers.

In Table 4, we compare LEXAM to prior legal QA datasets. We demonstrate that LEXAM a) covers a wide array of legal disciplines (78 subdomains), b) is the first general legal reasoning dataset containing challenging long-form ground-truth answers, and c) is the only dataset with an expert-tuned LLM-as-a-judge system for evaluation.

Because existing legal NLP benchmarks focus on narrow, task-specific formulations, whereas LEXAM evaluates holistic exam-style reasoning, direct aligned comparisons would require substantial reformatting and are not meaningful. We therefore provide descriptive comparisons and position LEXAM as complementary to these task-oriented datasets.

## 6.2 SWISS BENCHMARKS

Research on NLP benchmarks around Swiss legal data started with the task of legal judgment prediction, where models predict the case outcome based on facts (Niklaus et al., 2021; 2022). Subsequently, many other tasks have been studied on Swiss court rulings, such as explainability for judgment prediction (T.y.s.s. et al., 2024), negation scope resolution Christen et al. (2024), sentence boundary detection (Brugger et al., 2023), citation extraction (Rasiah et al., 2023), case importance prediction (Stern et al., 2024), law area prediction (Rasiah et al., 2023), summarization of leading decisions (Rolshoven et al., 2024), drafting of considerations (Rasiah et al., 2023), translation (Niklaus et al., 2025), and the analysis of gender distribution in court decisions (Merane, 2021). Additionally, prior work studied the anonymization (Niklaus et al., 2023a) and risk of re-identification of involved persons in court rulings (Nyffenegger et al., 2024).

Most prior tasks focus on content from court rulings and, to some extent, legislation. To our knowledge, no Swiss legal dataset from law school exams exists. In much prior work, labels have been extracted through regexes instead of manual human annotation (Merane & Geering, 2023). While regexes allow for very large-scale datasets, the covered tasks may not be as natural compared to fully human-annotated datasets. In this work, we collect a large dataset of real-world university law school exams, including detailed reference responses written by leading law professors.

## 7 CONCLUSION

We introduced LEXAM, a comprehensive legal reasoning benchmark composed of 340 law exams, specifically designed to challenge the performance of LLMs through both outcome- and process-based evaluations. Our analysis reveals substantial variability and limitations in LLM capabilities for addressing MCQs and especially on complex open questions; notably, increasing the number of MCQ options consistently reduces model accuracy. Our evaluation framework offers a scalable approach for assessing legal reasoning quality beyond simple accuracy metrics, thereby facilitating future research aimed at enhancing the reliability and robustness of LLMs on challenging legal tasks. Plans for further data collection are detailed in Appendix K.

## ETHICS STATEMENT

This work introduces a benchmark for evaluating LLMs on legal reasoning tasks using law school exam data. All exam questions and reference answers were obtained with permission from the University of Zurich Faculty of Law and are used exclusively for research purposes. The dataset consists solely of publicly available academic materials, namely professor-written exam questions and model solutions, and excludes any personal, confidential, or sensitive content, such as student-written answers.

Because the dataset contains no personal or human subject data, no anonymization procedures were necessary, and no institutional ethics board review was required under Swiss research regulations. This assessment was confirmed through consultation with multiple legal experts.

Our analyses focus exclusively on model performance and do not constitute legal advice. Moreover, while the benchmark reflects authentic legal reasoning tasks, it is intended purely as a research and educational resource, not as a tool for any real-world legal practice. Given the high-stakes nature of the legal domain, we strongly emphasize that current LLMs must not be relied upon for legal decision-making without qualified human oversight.

## REPRODUCIBILITY STATEMENT

To facilitate reproducibility and future research, we release both the full LEXAM dataset and the complete evaluation pipeline. Specifically, (a) the dataset—which includes all exam questions, metadata, and reference materials—and (b) the evaluation code are provided in the anonymous repository and uploaded as supplementary materials. Together, these resources enable replication of our results and support extensions of the benchmark to new models, tasks, and evaluation methods.

## ACKNOLEDGEMENT

We thank Benjamin Chen, Alexander Hoyle, Luka Nenadic, and Uriel Stettner for their very helpful comments throughout this research.

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

## A    USE OF AI ASSISTANTS

We used AI assistants, including GPT-4o, GPT-4.5, and GPT-5, for coding, shortening texts and editing LaTeX more efficiently. The AI tools are not directly used in writing, but assisting human through criticizing, grammar checks, etc..

## B    DATASET DETAILS

### B.1    INSTITUTIONAL BACKGROUND

Switzerland's legal system belongs to the civil law tradition and shares many characteristics with other continental European jurisdictions (though with the important distinction that Switzerland is not a member of the European Union). Legal interpretation in Switzerland focuses primarily on statutory texts. However, case law also plays a supporting role, especially decisions from higher courts, which contribute to legal clarity and consistency. Nevertheless, legal education in Switzerland emphasizes deductive reasoning (applying general legal rules to specific facts) rather than reliance on precedent.

Switzerland has several law faculties, with the University of Zurich hosting the largest. Legal education in Switzerland follows the Bologna model, consisting of a three-year Bachelor of Law (BLaw) and a two-year Master of Law (MLaw). The Bachelor program is typically highly structured, with a fixed curriculum covering the foundational areas of Swiss law. In contrast, the Master program allows for greater specialization and elective choices, enabling students to focus on specific legal domains. Upon graduation, students may pursue the bar exam, which is organized at the cantonal level. This exam places more emphasis on practical legal skills, such as drafting legal documents, applying procedural law, and demonstrating familiarity with cantonal and professional regulations.

### B.2    DATASET STATISTICS

**Open Questions (Test: 2,541, Dev: 300):** This set contains 2,541 questions (17.2% English, rest German), averaging 317.6 questions/year (2016-2023). Key distributions include: Private Law (43.5%), Public Law (38.4%), Criminal Law (9.0%), and interdisciplinary (9.1%); and Swiss jurisdiction (62.8%), international/non-Swiss (20.1%), and generic (17.1%). Average question/answer lengths are 174.3/246.6 words.

**Open Questions Dev (300 questions):** Sampled per course, this set of 300 questions mirrors the test set's distributions for legal area, jurisdiction, language, and question/answer length.

**MCQs (4,696 questions):** These questions are 62.8% German and 37.2% English. 91.5% target Swiss jurisdiction. Topics include: interdisciplinary (32%), Public Law (28.6%), Criminal Law (27.0%), and Private Law (11.5%). Questions average 3.7 choices. Unlike open questions, MCQ distribution by year is uneven, peaking in 2019 and 2023 (approx. 500 each year).

**MCQs Extended (385 questions):** All 385 questions target Swiss jurisdiction. Distributions are similar to the main MCQs: 64.4% German; Public Law (27.5%), Criminal Law (34.8%), Interdisciplinary (35.6%), with most samples from 2019 and 2023.

Table 5: Number of questions by Jurisdiction, Area, and Language

(a) Dev Dataset

| Jurisdiction | Criminal | | Interdisc. | | Private | | Public | | Total |
|---|---|---|---|---|---|---|---|---|---|
| | de | en | de | en | de | en | de | en | |
| Generic | 0 | 0 | 10 | 10 | 10 | 5 | 20 | 0 | 55 |
| International | 5 | 5 | 0 | 0 | 15 | 25 | 10 | 30 | 90 |
| Swiss | 20 | 0 | 0 | 0 | 80 | 5 | 50 | 0 | 155 |
| **Total** | **25** | **5** | **10** | **10** | **105** | **35** | **80** | **30** | **300** |

(b) Test Dataset

| Jurisdiction | Criminal | | Interdisc. | | Private | | Public | | Total |
|---|---|---|---|---|---|---|---|---|---|
| | de | en | de | en | de | en | de | en | |
| Generic | 0 | 0 | 193 | 37 | 20 | 21 | 164 | 0 | 435 |
| International | 18 | 26 | 0 | 0 | 56 | 211 | 67 | 132 | 510 |
| Swiss | 185 | 0 | 0 | 1 | 789 | 8 | 613 | 0 | 1596 |
| **Total** | **203** | **26** | **193** | **38** | **865** | **240** | **844** | **132** | **2541** |

Table 6: Number of records by Jurisdiction, Area, Course and Year on Open Questions (test)

| Jurisdiction | Area | Course | 2016 | 2017 | 2018 | 2019 | 2020 | 2021 | 2022 | 2023 | Total |
|---|---|---|---|---|---|---|---|---|---|---|---|
| Generic | Interdisciplinary | Juristische Zeitgeschichte | 6 | 0 | 0 | 0 | 0 | 0 | 0 | 0 | 6 |
| | | Legal Sociology | 0 | 0 | 0 | 6 | 0 | 0 | 4 | 0 | 10 |
| | | Legal Theory | 0 | 4 | 6 | 3 | 0 | 4 | 3 | 7 | 27 |
| | | Methodenlehre | 6 | 0 | 0 | 0 | 0 | 0 | 0 | 0 | 6 |
| | | Rechtsgeschichte | 10 | 14 | 18 | 14 | 0 | 0 | 0 | 0 | 56 |
| | | Wirtschaftsrechtsgeschichte | 16 | 21 | 16 | 19 | 15 | 12 | 12 | 14 | 125 |
| | Private | Antike Rechtsgeschichte | 10 | 0 | 0 | 0 | 0 | 0 | 0 | 0 | 10 |
| | | History of Business Law | 0 | 0 | 0 | 0 | 0 | 12 | 9 | 0 | 21 |
| | | Privatrechtsgeschichte | 10 | 0 | 0 | 0 | 0 | 0 | 0 | 0 | 10 |
| | Public | Gegenwartsprobleme der politischen Ethik | 0 | 0 | 4 | 0 | 0 | 0 | 0 | 0 | 4 |
| | | Kirchenrechtsgeschichte und Kirchenrecht | 4 | 13 | 17 | 8 | 0 | 9 | 20 | 13 | 84 |
| | | Recht und Religion | 0 | 0 | 0 | 0 | 0 | 10 | 0 | 14 | 24 |
| | | Rechtsphilosophie | 0 | 9 | 12 | 8 | 3 | 5 | 4 | 3 | 44 |
| | | Verfassungsgeschichte der Neuzeit | 0 | 8 | 0 | 0 | 0 | 0 | 0 | 0 | 8 |
| International | Criminal | International Criminal Law | 0 | 0 | 0 | 0 | 0 | 5 | 11 | 10 | 26 |
| | | Internationale Rechtshilfe in Strafsachen | 0 | 0 | 0 | 0 | 0 | 6 | 0 | 0 | 6 |
| | | Internationales und Europäisches Strafrecht | 0 | 4 | 0 | 4 | 4 | 0 | 0 | 0 | 12 |
| | Private | Comparative Corporate Law | 0 | 0 | 0 | 0 | 6 | 3 | 0 | 0 | 9 |
| | | Comparative Private Law | 0 | 0 | 0 | 11 | 0 | 7 | 6 | 0 | 24 |
| | | Europäisches Privatrecht | 4 | 6 | 0 | 0 | 0 | 0 | 0 | 0 | 10 |
| | | Foundations and Trusts | 0 | 0 | 0 | 0 | 0 | 6 | 13 | 0 | 19 |
| | | International Commercial Arbitration | 0 | 8 | 6 | 3 | 0 | 2 | 7 | 3 | 29 |
| | | International Sales Law | 0 | 0 | 0 | 0 | 0 | 5 | 16 | 0 | 21 |
| | | Internationales Privatrecht | 0 | 4 | 10 | 6 | 7 | 5 | 2 | 2 | 36 |
| | | Internationales Zivilverfahrensrecht | 0 | 1 | 3 | 2 | 0 | 4 | 0 | 0 | 10 |
| | | Introduction to Sports Law | 0 | 0 | 0 | 2 | 0 | 6 | 1 | 2 | 11 |
| | | Principles of Corporate Law | 0 | 0 | 0 | 0 | 0 | 5 | 0 | 6 | 11 |
| | | US Business Law | 0 | 0 | 0 | 18 | 0 | 0 | 62 | 7 | 87 |
| | Public | Comparative Constitutional Law | 0 | 0 | 0 | 0 | 0 | 0 | 0 | 9 | 9 |
| | | European Economic Law | 0 | 4 | 3 | 4 | 6 | 0 | 3 | 3 | 23 |
| | | International Economic Law | 0 | 4 | 4 | 0 | 0 | 0 | 0 | 0 | 8 |
| | | International Finance Law | 0 | 0 | 0 | 5 | 0 | 7 | 6 | 5 | 23 |
| | | International Financial Law | 3 | 4 | 4 | 0 | 0 | 0 | 0 | 0 | 11 |
| | | International Human Rights | 4 | 3 | 4 | 0 | 0 | 0 | 0 | 0 | 11 |
| | | International Organisations | 0 | 5 | 0 | 7 | 5 | 0 | 10 | 13 | 40 |
| | | Internationales Steuerrecht | 0 | 0 | 0 | 0 | 6 | 0 | 0 | 0 | 6 |
| | | Recht der Gewaltanwendung und Humanitäres Völkerrecht | 6 | 7 | 7 | 10 | 13 | 0 | 6 | 11 | 60 |
| | | Transnational Public Security Law | 8 | 0 | 0 | 0 | 0 | 0 | 0 | 0 | 8 |
| Swiss | Criminal | Jugendstrafrecht und Sanktionenrecht | 0 | 0 | 0 | 22 | 0 | 15 | 21 | 22 | 80 |
| | | Nebenstrafrecht | 0 | 1 | 9 | 3 | 0 | 4 | 9 | 21 | 47 |
| | | Strafrecht und Strafverfahrensrecht | 3 | 3 | 3 | 6 | 4 | 3 | 11 | 7 | 40 |
| | | Wirtschaftsstrafrecht | 0 | 2 | 0 | 4 | 9 | 0 | 0 | 3 | 18 |
| | Interdisciplinary | (Introduction to) Swiss Law | 0 | 0 | 0 | 1 | 0 | 0 | 0 | 0 | 1 |
| | Private | Aktienrecht | 0 | 0 | 0 | 9 | 5 | 0 | 7 | 0 | 21 |
| | | Alternative Streitbeteiligung | 0 | 0 | 0 | 4 | 0 | 0 | 9 | 9 | 22 |
| | | Arbeitsrecht | 10 | 9 | 8 | 10 | 9 | 9 | 0 | 0 | 55 |
| | | Bankrecht | 0 | 19 | 16 | 18 | 16 | 19 | 17 | 15 | 120 |
| | | Gesellschaftsrecht | 0 | 0 | 4 | 1 | 3 | 0 | 0 | 0 | 8 |
| | | Grundbuchrecht | 0 | 7 | 0 | 5 | 0 | 8 | 0 | 7 | 27 |
| | | Güter- und Erbrecht | 0 | 0 | 0 | 0 | 0 | 0 | 10 | 7 | 17 |
| | | Haftpflicht- und Versicherungsrecht | 10 | 11 | 0 | 12 | 12 | 7 | 3 | 5 | 60 |
| | | Handels- und Wirtschaftsrecht | 1 | 0 | 0 | 0 | 0 | 0 | 12 | 4 | 17 |
| | | Immaterialgüterrecht | 0 | 0 | 0 | 0 | 0 | 0 | 5 | 7 | 12 |
| | | Immobiliarsachenrecht | 7 | 0 | 5 | 3 | 8 | 9 | 7 | 10 | 49 |
| | | Informations- und Kommunikationsrecht | 0 | 2 | 8 | 8 | 8 | 0 | 0 | 0 | 26 |
| | | Kapitalmarktrecht | 5 | 5 | 7 | 5 | 5 | 6 | 2 | 7 | 42 |
| | | Kolloquium zum Allgemeinen Teil des Obligationen rechts | 0 | 0 | 0 | 6 | 5 | 0 | 0 | 0 | 11 |
| | | Kunst- und Kulturrecht | 14 | 0 | 9 | 0 | 11 | 12 | 18 | 0 | 64 |
| | | Lizenzvertrags- und Lizenzkartellrecht | 25 | 22 | 10 | 20 | 6 | 14 | 0 | 0 | 97 |
| | | Nachlassplanung | 4 | 3 | 5 | 8 | 15 | 3 | 11 | 7 | 56 |
| | | Notariatsrecht | 0 | 0 | 8 | 0 | 6 | 0 | 8 | 0 | 22 |
| | | Scheidungsrecht/Partnerschaftsauflösung | 0 | 3 | 4 | 0 | 0 | 0 | 0 | 0 | 7 |
| | | Wertpapierrecht | 0 | 5 | 0 | 0 | 0 | 0 | 0 | 0 | 5 |
| | | Zivilverfahrensrecht | 5 | 2 | 4 | 4 | 3 | 6 | 9 | 26 | 59 |
| | Public | Bundesverwaltungsrecht | 0 | 4 | 5 | 0 | 0 | 0 | 0 | 0 | 9 |
| | | Demokratie | 0 | 0 | 5 | 0 | 0 | 0 | 0 | 0 | 5 |
| | | Gesundheitsrecht und Bioethik | 0 | 0 | 0 | 0 | 6 | 0 | 0 | 0 | 6 |
| | | Migrationsrecht | 0 | 20 | 15 | 24 | 19 | 25 | 14 | 0 | 117 |
| | | Raumplanungs- und Baurecht | 6 | 5 | 5 | 6 | 8 | 10 | 0 | 5 | 45 |
| | | Rechtsetzungslehre | 14 | 0 | 0 | 21 | 0 | 4 | 13 | 19 | 71 |
| | | Sicherheits-, Polizei-, und Menschenrechte | 7 | 9 | 7 | 9 | 12 | 5 | 10 | 9 | 68 |
| | | Sozialversicherungsrecht | 0 | 10 | 16 | 14 | 12 | 12 | 13 | 14 | 91 |
| | | Steuerrecht | 0 | 6 | 6 | 4 | 9 | 7 | 10 | 16 | 58 |
| | | Umweltrecht | 0 | 4 | 4 | 7 | 11 | 6 | 0 | 0 | 32 |
| | | Unternehmenssteuerrecht | 0 | 12 | 12 | 11 | 20 | 0 | 0 | 8 | 63 |
| | | Öffentliches Verfahrensrecht | 6 | 6 | 7 | 8 | 8 | 5 | 8 | 0 | 48 |
| **Total** | | | **204** | **289** | **296** | **383** | **295** | **302** | **430** | **342** | **2541** |

Table 7: Number of records by Jurisdiction, Area, Course and Year on Open Questions (dev)

| Jurisdiction | Area | Course | 2016 | 2017 | 2018 | 2019 | 2020 | 2021 | 2022 | 2023 | Total |
|---|---|---|---|---|---|---|---|---|---|---|---|---|
| Generic | Interdisciplinary | Legal Sociology | 0 | 0 | 0 | 4 | 0 | 0 | 1 | 0 | 5 |
| | | Legal Theory | 0 | 0 | 2 | 1 | 0 | 1 | 1 | 0 | 5 |
| | | Rechtsgeschichte | 0 | 1 | 1 | 3 | 0 | 0 | 0 | 0 | 5 |
| | | Wirtschaftsrechtsgeschichte | 0 | 0 | 1 | 0 | 1 | 0 | 2 | 1 | 5 |
| | Private | Antike Rechtsgeschichte | 5 | 0 | 0 | 0 | 0 | 0 | 0 | 0 | 5 |
| | | History of Business Law | 0 | 0 | 0 | 0 | 0 | 2 | 3 | 0 | 5 |
| | | Privatrechtsgeschichte | 5 | 0 | 0 | 0 | 0 | 0 | 0 | 0 | 5 |
| | Public | Kirchenrechtsgeschichte und Kirchenrecht | 0 | 0 | 0 | 3 | 0 | 0 | 1 | 1 | 5 |
| | | Recht und Religion | 0 | 0 | 0 | 0 | 0 | 2 | 0 | 3 | 5 |
| | | Rechtsphilosophie | 0 | 1 | 0 | 3 | 1 | 0 | 0 | 0 | 5 |
| | | Verfassungsgeschichte der Neuzeit | 0 | 5 | 0 | 0 | 0 | 0 | 0 | 0 | 5 |
| International | Criminal | International Criminal Law | 0 | 0 | 0 | 0 | 0 | 0 | 2 | 3 | 5 |
| | | Internationales und Europäisches Strafrecht | 0 | 3 | 0 | 1 | 1 | 0 | 0 | 0 | 5 |
| | Private | Comparative Private Law | 0 | 0 | 0 | 2 | 0 | 2 | 1 | 0 | 5 |
| | | Europäisches Privatrecht | 1 | 4 | 0 | 0 | 0 | 0 | 0 | 0 | 5 |
| | | Foundations and Trusts | 0 | 0 | 0 | 0 | 0 | 2 | 3 | 0 | 5 |
| | | International Commercial Arbitration | 0 | 2 | 2 | 0 | 0 | 1 | 0 | 0 | 5 |
| | | International Sales Law | 0 | 0 | 0 | 0 | 0 | 2 | 3 | 0 | 5 |
| | | Internationales Privatrecht | 0 | 1 | 1 | 0 | 0 | 1 | 1 | 1 | 5 |
| | | Internationales Zivilverfahrensrecht | 0 | 1 | 1 | 2 | 0 | 1 | 0 | 0 | 5 |
| | | US Business Law | 0 | 0 | 0 | 2 | 0 | 0 | 2 | 1 | 5 |
| | Public | European Economic Law | 0 | 0 | 0 | 0 | 1 | 0 | 2 | 2 | 5 |
| | | International Economic Law | 0 | 2 | 3 | 0 | 0 | 0 | 0 | 0 | 5 |
| | | International Finance Law | 0 | 0 | 0 | 1 | 0 | 1 | 2 | 1 | 5 |
| | | International Financial Law | 1 | 2 | 2 | 0 | 0 | 0 | 0 | 0 | 5 |
| | | International Human Rights | 2 | 1 | 2 | 0 | 0 | 0 | 0 | 0 | 5 |
| | | International Organisations | 0 | 0 | 0 | 0 | 0 | 0 | 3 | 2 | 5 |
| | | Internationales Steuerrecht | 0 | 0 | 0 | 0 | 5 | 0 | 0 | 0 | 5 |
| | | Recht der Gewaltanwendung und Humanitäres Völkerrecht | 1 | 0 | 0 | 1 | 0 | 0 | 2 | 1 | 5 |
| Swiss | Criminal | Jugendstrafrecht und Sanktionenrecht | 0 | 0 | 0 | 0 | 0 | 0 | 2 | 3 | 5 |
| | | Nebenstrafrecht | 0 | 0 | 2 | 1 | 0 | 1 | 0 | 1 | 5 |
| | | Strafrecht und Strafverfahrensrecht | 0 | 0 | 0 | 0 | 1 | 0 | 1 | 3 | 5 |
| | | Wirtschaftsstrafrecht | 0 | 0 | 0 | 0 | 3 | 0 | 0 | 2 | 5 |
| | Private | Aktienrecht | 0 | 0 | 0 | 2 | 2 | 0 | 1 | 0 | 5 |
| | | Alternative Streitbeteiligung | 0 | 0 | 0 | 0 | 0 | 0 | 2 | 3 | 5 |
| | | Arbeitsrecht | 0 | 2 | 1 | 2 | 0 | 0 | 0 | 0 | 5 |
| | | Bankrecht | 0 | 0 | 3 | 0 | 1 | 0 | 0 | 1 | 5 |
| | | Gesellschaftsrecht | 0 | 0 | 1 | 2 | 2 | 0 | 0 | 0 | 5 |
| | | Grundbuchrecht | 0 | 2 | 0 | 1 | 0 | 2 | 0 | 0 | 5 |
| | | Güter- und Erbrecht | 0 | 0 | 0 | 0 | 0 | 0 | 3 | 2 | 5 |
| | | Haftpflicht- und Versicherungsrecht | 1 | 0 | 0 | 1 | 1 | 0 | 1 | 1 | 5 |
| | | Immobiliarsachenrecht | 0 | 0 | 3 | 0 | 0 | 1 | 1 | 0 | 5 |
| | | Informations- und Kommunikationsrecht | 0 | 0 | 2 | 0 | 3 | 0 | 0 | 0 | 5 |
| | | Kapitalmarktrecht | 2 | 1 | 2 | 0 | 0 | 0 | 0 | 0 | 5 |
| | | Kolloquium zum Allgemeinen Teil des Obligationenrechts | 0 | 0 | 0 | 2 | 3 | 0 | 0 | 0 | 5 |
| | | Kunst- und Kulturrecht | 1 | 0 | 3 | 0 | 0 | 1 | 0 | 0 | 5 |
| | | Lizenzvertrags- und Lizenzkartellrecht | 1 | 0 | 2 | 1 | 1 | 0 | 0 | 0 | 5 |
| | | Nachlassplanung | 0 | 0 | 0 | 0 | 2 | 1 | 1 | 1 | 5 |
| | | Notariatsrecht | 0 | 0 | 1 | 0 | 1 | 0 | 3 | 0 | 5 |
| | | Zivilverfahrensrecht | 1 | 1 | 0 | 0 | 0 | 1 | 0 | 2 | 5 |
| | Public | Bundesverwaltungsrecht | 0 | 1 | 4 | 0 | 0 | 0 | 0 | 0 | 5 |
| | | Migrationsrecht | 0 | 0 | 0 | 0 | 3 | 1 | 1 | 0 | 5 |
| | | Raumplanungs- und Baurecht | 2 | 0 | 2 | 1 | 0 | 0 | 0 | 0 | 5 |
| | | Rechtsetzungslehre | 1 | 0 | 0 | 1 | 0 | 0 | 1 | 2 | 5 |
| | | Sicherheits-, Polizei-, und Menschenrechte | 0 | 0 | 3 | 1 | 0 | 0 | 1 | 0 | 5 |
| | | Sozialversicherungsrecht | 0 | 1 | 0 | 0 | 2 | 1 | 1 | 0 | 5 |
| | | Steuerrecht | 0 | 1 | 0 | 0 | 2 | 1 | 0 | 1 | 5 |
| | | Umweltrecht | 0 | 2 | 1 | 1 | 1 | 0 | 0 | 0 | 5 |
| | | Unternehmenssteuerrecht | 0 | 1 | 1 | 0 | 1 | 0 | 2 | 0 | 5 |
| | | Öffentliches Verfahrensrecht | 2 | 1 | 2 | 0 | 0 | 0 | 0 | 0 | 5 |
| **Grand Total** | | | **26** | **36** | **48** | **39** | **38** | **25** | **50** | **38** | **300** |

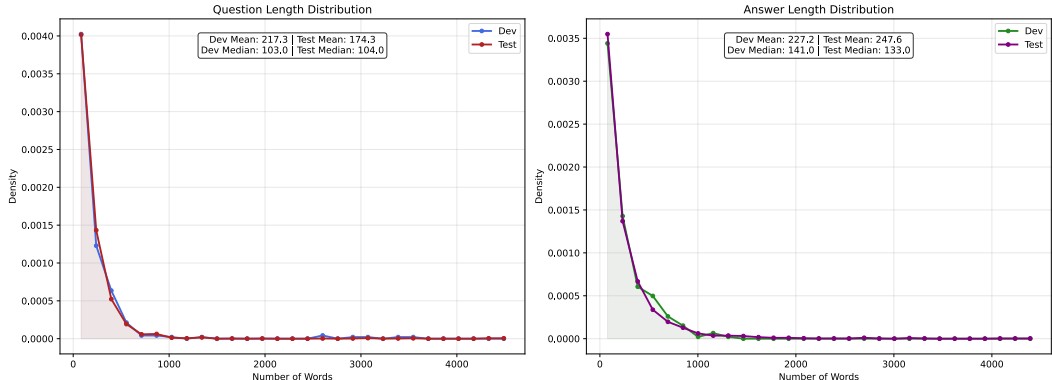

Figure 6: Distribution of the number of words per question and answer in the open-ended question dataset.

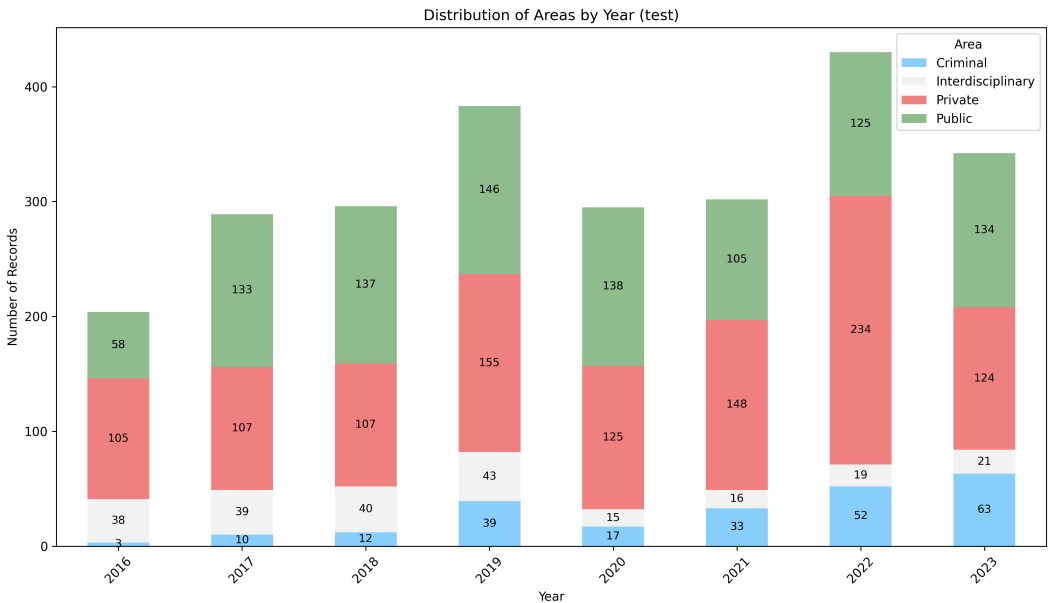

Figure 7: Yearly distribution of the number of question in the Open question test dataset, per legal area.

## C  SUPPLEMENTARY RESULTS ON MODEL PERFORMANCE

### C.1  OPEN QUESTIONS

This section presents additional analyses to complement the main results in Figure 4, including *Challenging Course*, *Top vs. Bottom Courses*, *Year*, *Question Length* and *Answer Length*. The feature *Challenging Course* represents courses taught in German, focusing on Swiss law, and lacking an interdisciplinary approach. Following a manual review conducted by legal experts from the author team, this feature serves as a proxy for legally intensive and advanced-level courses, which, while not all directly relevant to the bar exam, generally include fewer introductory courses and exhibit higher overall difficulty. The *Top vs. Bottom Courses* feature is constructed by computing the mean model score for each course and identifying the ten courses with the highest and lowest average performance, respectively. This allows for a contrastive view of where models excel versus where they struggle most.

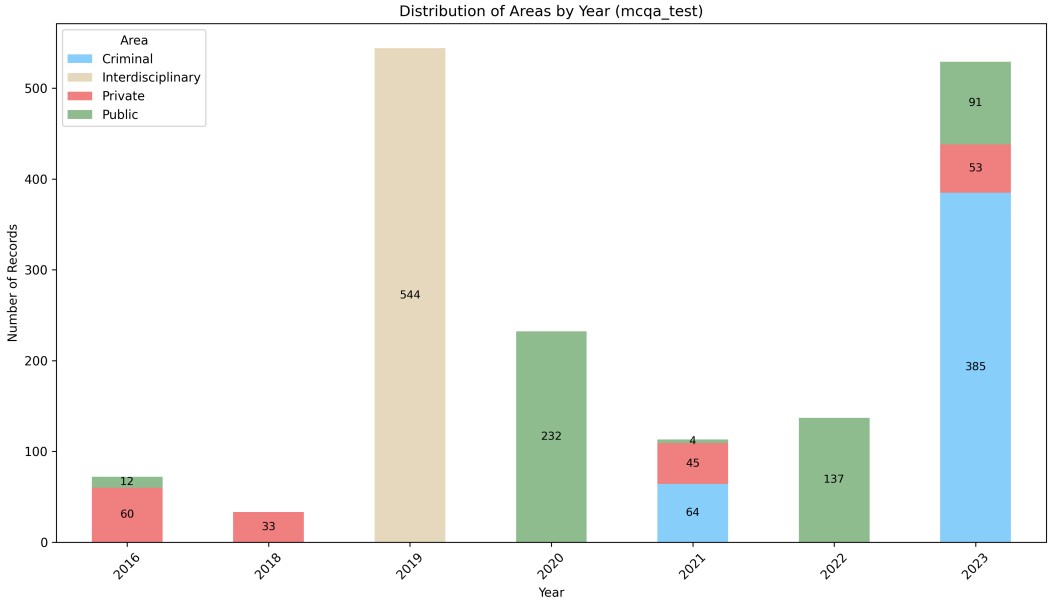

Figure 8: Yearly distribution of the number of question in the MCQs dataset, per legal area.

Table 8: Number of the 385 MCQs with perturbations per course.

| Course | # Questions |
|---|---|
| (Introduction to) Swiss Law | 137 |
| Nebenstrafrecht | 58 |
| Steuerrecht | 56 |
| Unternehmenssteuerrecht | 33 |
| Strafrecht und Strafverfahrensrecht | 32 |
| Jugendstrafrecht und Sanktionenrecht | 24 |
| Wirtschaftsstrafrecht | 20 |
| Rechtsetzungslehre | 16 |
| Lizenzvertrags- und Lizenzkartellrecht | 7 |
| Raumplanungs- und Baurecht | 1 |
| Nachlassplanung | 1 |

For *Challenging Course* (D) and *Top vs. Bottom Courses* (E), we observe a similar pattern to the main features, where reasoning models outperform large and small models. Additionally, accuracy is consistently higher for non-challenging courses, and all three model groups achieve significantly higher scores on the top 10 courses, while struggling on the bottom 10.

The feature *Year* (F) tracks model performance over time from 2016 to 2023, where lines indicate the average LLM-Judge score, and the shaded areas represent the bootstrap standard error. Across all three model groups, there is a clear downward trend in performance on more recent questions, particularly from 2019 onward, suggesting that newer courses or exams may involve harder or more nuanced legal content. The decline is most pronounced for small models, followed by reasoning models, while large models exhibit comparatively more stable performance over time. Although reasoning models also experience this decline, they still maintain a consistent advantage over the other groups in all years. However, we note that course availability and composition vary across years. Not all courses are offered every year, and the distribution of features such as language, jurisdiction, or difficulty may shift. Accordingly, part of the performance variation over time may reflect changes in the underlying course set, rather than purely model-related effects.

Regarding the impact of input length on model performance, we apply separate x-axis limits for the two plots due to the skewed distribution of question and answer lengths. *Question Length* (G)

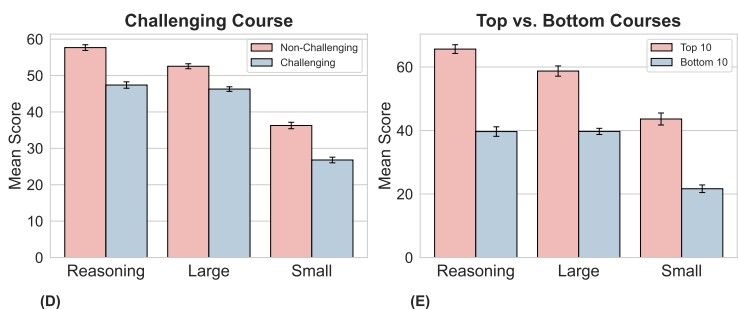

Figure 9: Model performance on open questions grouped by extended features

is capped at 400 words and *Answer Length* (H) at 600 words, covering approximately 90% of the data in each case. We observe that model performance is relatively stable in all model groups by question length, with a slight upward trend as questions become longer. In contrast, performance tends to decrease with increasing answer length. The results may indicate that longer questions provide additional context or clarification that benefits model reasoning, but longer outputs may introduce complexity for solving problems for these models. Among the model groups, reasoning models maintain higher and more stable accuracy in both conditions.

## C.2 MULTIPLE-CHOICE QUESTIONS

Table 9: Performance of models on 16-choice questions (MCQs) with bootstrapped standard error. Results are sorted by accuracy in descending order within each group.

|  | Model | MCQs Accuracy (± S.E) |
|---|---|---|
| Reasoning | GPT-5.2 | **52.53 (± 1.53)** |
| | Claude-4.6-sonnet | 52.43 (± 1.56) |
| | Qwen3.5-plus | 43.77 (± 1.57) |
| | DeepSeek-V3.2-reasoning | 41.15 (± 1.49) |
| | GPT-5-mini | 40.95 (± 1.52) |
| | GLM-5 | 39.01 (± 1.53) |
| | Kimi-k2.5 | 35.70 (± 1.52) |
| | MiniMax-m2.5 | 32.10 (± 1.51) |
| | Qwen3-32B | 30.25 (± 1.43) |
| | GPT-OSS-120B | 29.67 (± 1.41) |
| | GPT-5-nano | 28.89 (± 1.44) |
| | Qwen3-14B | 26.26 (± 1.36) |
| | Qwen3-8B | 24.81 (± 1.31) |
| | GPT-OSS-20B | 22.67 (± 1.28) |
| Large | Llama-3.3-70B-it | **26.07 (± 1.41)** |
| | Apertus-70B | 11.67 (± 1.00) |
| Small | Phi-4 | **21.40 (± 1.28)** |
| | Gemma-3-12B-it | 14.88 (± 1.10) |
| | Gemma-3-27B-it | 14.59 (± 1.10) |
| | Llama-3.1-8B-it | 12.35 (± 1.02) |
| | Apertus-8B | 7.49 (± 0.82) |

**MCQ-4.** As shown in Table 10, reasoning models lead in accuracy, with GPT-5 (62.65%) and Claude-4.5-Sonnet (58.01%) achieving the top two scores. These are closely followed by the strongest large model, GPT-4.1 (54.40%), which also demonstrates competitive performance. Among small models, GPT-4.1-mini (48.49%) and GPT-4o-mini (40.96%) outperform the rest, while most others in this category, including Gemma-2-9B-it, EuroLLM-9B-it, and Llama-3.1-8B-it, score below 30%. Across

all groups, bootstrap standard errors are low (typically below 1.3%), indicating stable performance estimates.

Table 10: Performance of models on 4-choice questions (MCQs) with bootstrapped standard error. Results are sorted by accuracy in descending order within each group.

| | Model | MCQs Accuracy (± S.E) |
|---|---|---|
| Reasoning | GPT-5 | **62.65 (± 1.17)** |
| | Claude-4.5-Sonnet | 58.01 (± 1.17) |
| | Claude-3.7-Sonnet | 57.23 (± 1.21) |
| | Gemini-2.5-Pro | 55.72 (± 1.18) |
| | GPT-5-mini | 54.82 (± 1.19) |
| | DeepSeek-V3.2-Exp | 53.07 (± 1.22) |
| | DeepSeek-R1 | 52.41 (± 1.22) |
| | Qwen3-235B | 48.19 (± 1.20) |
| | QwQ-32B | 47.83 (± 1.23) |
| | GPT-OSS-120B | 47.71 (± 1.21) |
| | GPT-5-nano | 47.11 (± 1.19) |
| | Qwen3-32B | 45.30 (± 1.23) |
| | O3-mini | 44.22 (± 1.23) |
| | Qwen3-Next | 43.31 (± 1.21) |
| | GPT-OSS-20B | 40.78 (± 1.23) |
| Large | GPT-4.1 | **54.40 (± 1.26)** |
| | GPT-4o | 53.13 (± 1.20) |
| | Llama-4-Maverick | 49.10 (± 1.24) |
| | DeepSeek-V3 | 46.57 (± 1.28) |
| | Llama-3.1-405B-it | 43.19 (± 1.19) |
| | Llama-3.3-70B-it | 28.19 (± 1.10) |
| Small | GPT-4.1-mini | **48.49 (± 1.22)** |
| | GPT-4o-mini | 40.96 (± 1.21) |
| | Phi-4 | 40.66 (± 1.19) |
| | GPT-4.1-nano | 39.22 (± 1.22) |
| | Gemma-3-12B-it | 29.94 (± 1.10) |
| | Qwen-2.5-7B-it | 29.28 (± 1.10) |
| | Ministral-8B-it | 26.27 (± 1.12) |
| | Gemma-2-9B-it | 25.36 (± 1.04) |
| | Llama-3.1-8B-it | 24.04 (± 1.05) |
| | Apertus-8B | 6.42 (± 1.02) |

Figure 10 demonstrates further results on model performance on multiple-choice questions related to the features *Challenging Course*, *None Option Included*, *Top vs. Bottom Courses*, *Year*, and *Question Length*, as an extension to Figure 5. We constructed the feature *None Option Included* to indicate whether or not "None of the statements (Keine der Aussagen)" appears as an answer choice, in order to examine whether LLMs can select correct answers under this condition.

In general, models perform worse on courses labeled as "challenging" (E). When the answer set includes a "none" option, performance declines modestly for all groups, likely due to increased interpretive ambiguity (F). Regarding the model performance by year (H), reasoning models have a consistent lead, but there is a general decline in accuracy across all model groups starting from 2018, which may reflect increasing difficulty or complexity in more recent multiple-choice questions. Notably, the gap between reasoning and non-reasoning models remains relatively stable, suggesting that architectural advantages in reasoning models persist even as task difficulty rises. In addition, accuracy does not consistently decline as the number of legal statements increases. Reasoning models maintain or even improve their performance on prompts with more statements, suggesting they benefit from richer contextual information. While large and small models show more variability, their accuracy also does not systematically drop, indicating that the length of the prompt alone may not be a key limiting factor. Bootstrapped standard errors are shaded around each trend line.

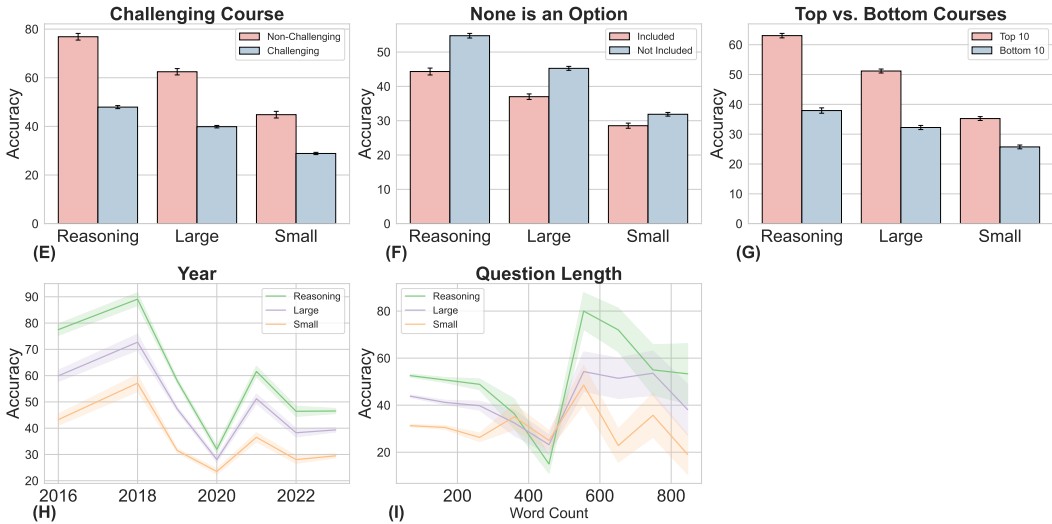

Figure 10: Model performance on MCQ grouped by extended features

Last, model performance across all model groups shows a non-monotonic relationship with question length (I). Accuracy decreases as questions grow longer, reaching a local minimum around 450 words, which may reflect increasing logical complexity that challenges model comprehension. Notably, performance rebounds significantly beyond this point, particularly for reasoning models, which achieve an accuracy increase above around 500 words. This suggests that very long questions may provide richer context or clearer constraints, making it easier for models, particularly reasoning models, to identify the correct answer.

## D    FURTHER ANALYSES ON MCQS WITH PERTURBATIONS

We provide additional results here on MCQs with perturbations. As shown in Figure 11, across all tested settings, the models exhibit substantial variations in performance, further validating the dataset's capability to effectively differentiate between models of varying abilities. In particular, Gemini-2.5-Pro consistently achieves the highest accuracy, maintaining a lead over the other models, while Claude-3.7-Sonnet consistently ranks second across all settings. However, all models experience a substantial drop in performance as the number of choices increases from 4 to 32. For example, the accuracy of Gemini-2.5-Pro drops from 68.6% at 4 choices to just 35.6% at 32 choices, nearly a 50% relative decrease. Similar steep declines are observed in other models: Sonnet drops from 60.9% to 33%, DeepSeek-R1 from 57.5% to 24.9%, and DeepSeek-V3 from 58.6% to 16%. The degradation is most pronounced in DeepSeek-V3, which loses over 40 percentage points in accuracy, suggesting significant difficulty with increased distractor complexity.

### D.1    PERFORMANCE PER COURSE

To further understand model behavior under varying levels of difficulty, Figures 12 present accuracy per course for each setting: 4, 8, 16, and 32 choices.

In the 4-choice setting, Gemini shows consistently high performance across nearly all legal sub-domains, achieving above 75% in half of the courses and peaking at at 87.5% in "Strafrecht und Strafverfahrensrecht (criminal law and criminal procedure law)." Sonnet and GPT-4.1 also perform relatively well in foundational areas such as "(Introduction to) Swiss law", but begin to show variability in more specialized domains, e.g. "Steuerrecht (tax law)". DeepSeek-V3 displays a spike in "Jugendstrafrecht und Sanktionenrecht (juvenile criminal law and sanctions law)", but is inconsistent in other areas. O3-mini ranks the lowest in almost all areas.

As the choice count increases to 8, a noticeable drop in accuracy is observed across all models, indicating their sensitivity to simple perturbations. Courses such as "Strafrecht und Strafverfahrensrecht

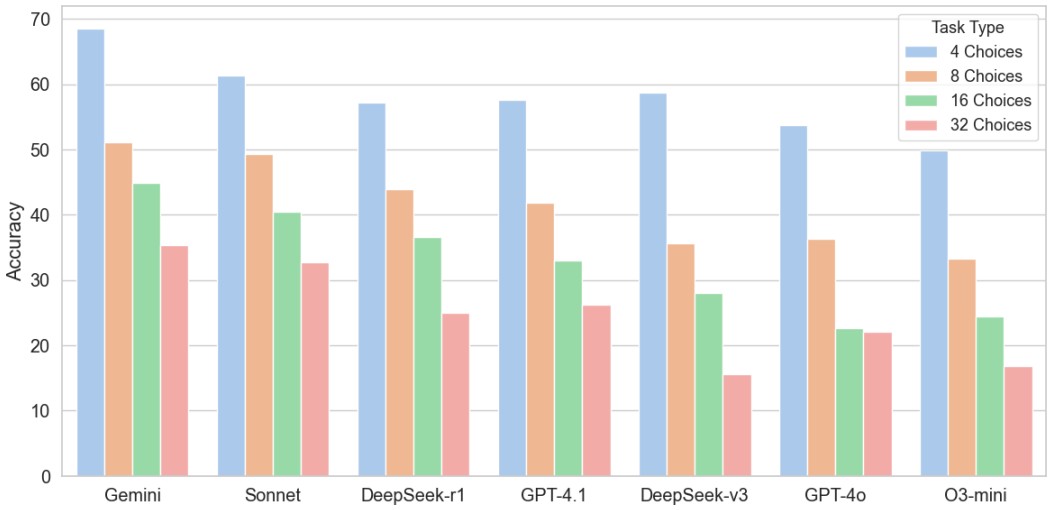

Figure 11: Model accuracy across 4, 8, 16, and 32 choices tasks

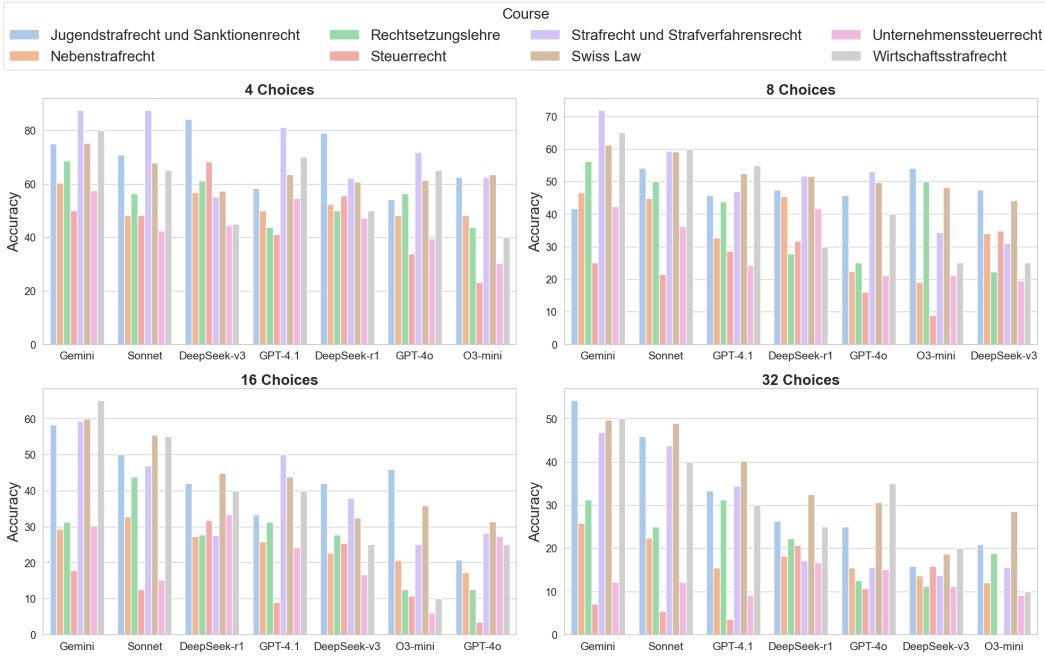

Figure 12: Model accuracy across 4, 8, 16, and 32 choices tasks, grouped by course

(criminal law and criminal procedure law)" and "(Introduction to) Swiss Law" remain relatively robust, whereas performance declines sharply in more technical or niche subjects, such as Steuerrecht (tax law). This trend continues at 16 choices, with many models dropping below 50% in most courses. By the 32-choice setting, accuracy is uniformly low across all models and courses. Even top-performing models like Gemini and Sonnet fail to maintain reliable performance in this highly perturbed condition. The gap between domains also widens, highlighting increasing domain sensitivity and difficulty.

Such per-course and per-setting analyses underscore the escalating challenges that an increasing number of distractor options pose to current language models, both reasoning and non-reasoning. They also highlight the importance of comprehensive domain coverage and robust generalization capacities.

## D.2 PERFORMANCE PER LANGUAGE

In addition to the course-wise breakdowns, we examined performance across languages, as shown in Figure 13. Across all difficulty levels, models perform better in English than in German, with the performance gap widening as complexity increases. For instance, at 4 choices, Gemini achieves 75.2% accuracy in English versus 64.9% in German, with a modest gap. However, by 32 choices, Gemini's accuracy drops to 49.6% in English and just 27.6% in German.

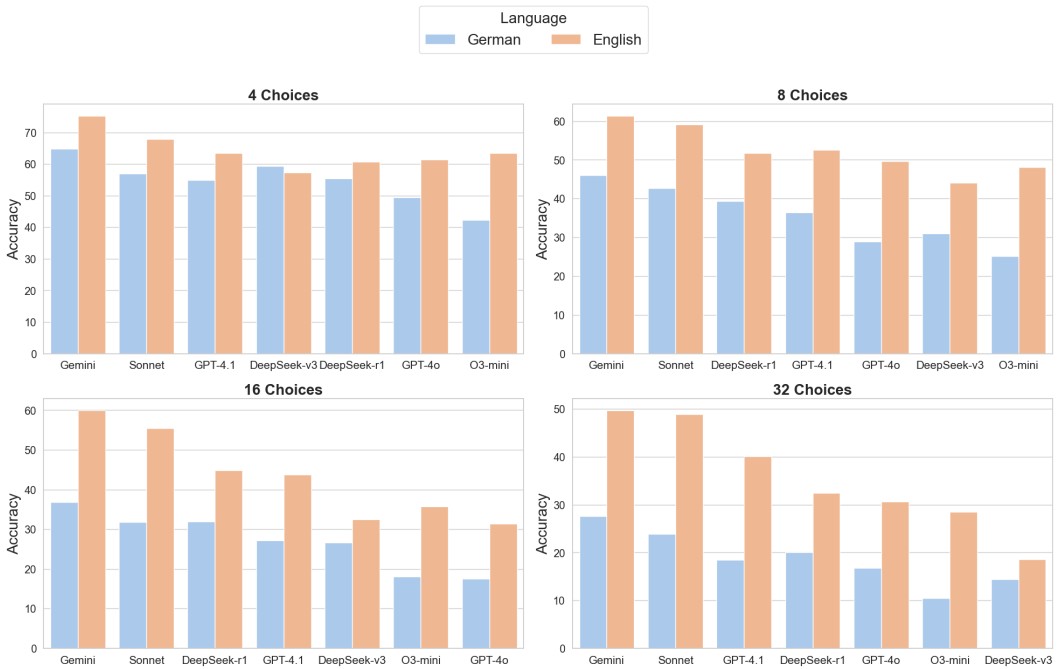

Figure 13: Model accuracy across 4, 8, 16, and 32 choices tasks, grouped by language

These findings suggest that while models demonstrate reasonable competence in multilingual scenarios at lower difficulty levels, they struggle significantly under more complex, high-choice conditions. The results underscore the need for improved multilingual training and evaluation to support the development of robust LLMs.

## D.3 PERFORMANCE PER LEGAL AREA

We further analyzed model performance across legal areas, as illustrated in Figure 14. Models consistently perform best on interdisciplinary law questions and worst on public law. This trend persists across all difficulty levels. For example, at the 4-choice level, Gemini scores 75.2% in interdisciplinary, 72.4% in criminal, and just 55.2% in public law. At the most challenging 32-choice level, its performance drops to 49.6% in interdisciplinary, 39.6% in criminal, and only 12.4% in public law. The steady performance advantage in interdisciplinary questions may reflect model strengths in synthesizing general legal knowledge, whereas public law questions, often more context-specific, present a greater challenge. These results indicate that the legal area substantially influences LLM performance, with public law emerging as a particularly challenging domain that may benefit from domain-specific training enhancements.

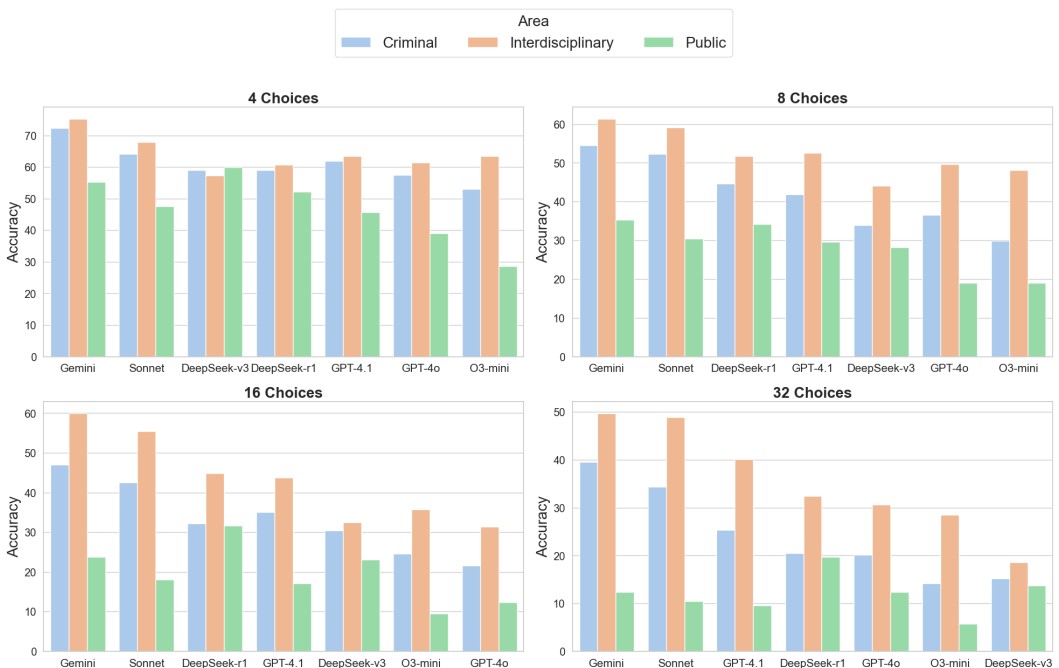

Figure 14: Model accuracy across 4, 8, 16, and 32 choices tasks, grouped by legal area

# E  PROMPT DESIGN

## E.1  PROMPT TO ANSWER OPEN QUESTIONS

You are an expert in {course_name} and address legal issues in a structured, exam-style manner.
Assume Swiss law applies unless specifically mentioned; if the course context justifies, address legal issues beyond Swiss law as well.
Use precise legal language and formal "Sie" when answering.
Do NOT state any disclaimer or refer to the need for external legal advice.
Do NOT request the user to consult laws or to research on their own.
Offer focused legal analyses and individualized advice.
Speak directly and authoritatively without mentioning that your response is merely for general information.
Incorporate Swiss-specific legal terminology.
If you have discovered relevant legal considerations (Erwägungen), respond with a concise, clear legal analysis.
Cite only from your identified considerations.
Always cite the specific legal provision, explicitly indicating paragraphs (Abs.), numbers (Ziff.), or letters (lit.) where available (e.g., "'Art. 74 Abs. 2 Ziff. 2 OR", "Art. 336 lit. a StGB"). Avoid general references (such as 'Art. 3 ZGB') without mentioning the specific paragraph, number, or letter, if applicable.
If no relevant considerations are found, explicitly state that no pertinent information is available.
If you do have reliable sources, share practical guidance or insights from them.
Respond in the same language as the question.
If the question specifically requests a short answer, provide a concise response.
If the prompt asks you to analyze a specific case provided in the exam, but the text or details of that case have not been provided in the prompt, explicitly flag that the required case material is missing.

Question:
"'{question}'"

Answer:

### E.2 PROMPT TO JUDGE OPEN QUESTIONS ANSWERS

The judging prompt follows a bottom-up design grounded in the professor-written model answers that reflect standard law-school reasoning patterns (issue identification, rule selection, application, and treatment of alternatives). It was iteratively refined by two authors with doctoral-level legal training, ensuring that the evaluation captures domain-specific expectations such as penalizing fabricated statutory references and respecting legally permissible solution paths. These elements distinguish legal-reasoning evaluation from general-reasoning evaluation.

---

**System Prompt**
Act as a Judge specializing in the evaluation of Swiss law schools exams. Your task is to assess how well the response aligns with the reference answer, with a focus on accuracy, completeness, and legal reasoning.
**User Prompt**
Goal: Your task is to assess how well the response aligns with the reference answer, with a focus on accuracy, completeness, and legal reasoning.

Context: You will be provided with a response (labeled: Model's Answer) to a law school exam question (labeled: Question) and a reference answer (labeled: Reference Answer).

Return format:
After reviewing the response:
1. Explanation: Briefly explain your reasoning regarding how the response conforms to or deviates from the reference answer.
2. Constructive feedback: Additionally, provide neutral, constructive feedback and corrections in the style of a university professor.
3. Correctness score: Assign a final correctness score on a scale from 0.0 to 1.0 (in increments of 0.1). This score should reflect the extent to which the response satisfies the reference answer, where
- 1.0 = complete fulfillment (100- lower scores reflect proportionate shortfalls (e.g. 0.5 = 50% fulfillment).
- strictly follow the format: [[score]], e.g., The correctness score: [[0.5]]..

Warnings:
- In some cases, the reference answer may include only keywords or factual elements to be examined, along with (+), (-) or (+/-). Respect these indications when determining correctness:
- (+) means the element must be affirmed.
- (–) means the element must be denied.
- (-/+) indicates that arguments in either direction are acceptable if legally sound.
- Deviations or additional elements not found in the reference answer should generally be penalized unless you are certain they are legally correct and relevant. Assume the reference answer includes all information necessary for a perfect response.
- The reference answer may contain citations (e.g., from books or law review articles), which the response does not need to replicate. However, statutes should be cited precisely, specifying Abs., Ziff., or lit. whenever applicable.
- If the reference answer includes separate sub-points, use these for proportional scoring guidance (e.g., addressing 2 out of 4 sub-points correctly equals approximately a 0.5 score).
Judge the below case, give the brief reasoning process and the final grade.

Question:
"'{question}'"

Reference Answer:
"'{reference_answer}'"

Model's Answer:
"'{model_answer}'"

Your Judgment:

---

### E.3 SAMPLE OPEN QUESTION

**Question**:
A (domiciled in Switzerland), B (domiciled in Switzerland) and C (domiciled in France) enter into a contract. The contract contains an arbitration clause that provides as follows: "Any dispute arising out of or in connection with this agreement shall be resolved either by arbitration in Switzerland or by French courts." Is the PILA applicable to determine the validity of the above "arbitration agreement" in case B intends to initiate arbitration proceedings against A?

**Reference Answer**:
Chapter 12 PILA: Art 176 (1) PILA
1. seat of tribunal in CH?
- can be determined by the parties
- "seat" means, in principle, place, not a country
-> here: "seat in Switzerland" -> no specific seat chosen
- majority view: could lead to the invalidity of the AA (see Poudret/Besson, n. 139)
- Interpretation of this wording? (according to the interpretation of contracts)
- Art. 355 (2) CCP per analogiam? (disputed)
- Art. 176 (3) PILA: seat can be determined by the tribunal or the institution
-> in casu: no institutional arbitration, and no arbitral proceedings initiated
-> arbitrators must be appointed, or at least Canton/juge d'appui specified (Art. 179 PILA)
2. domicile of at least one party not in CH?
- when? -> at the time of the conclusion of the AA
- Parties to the contract or parties to the proceedings?
-> according to the SFT (see 4P.54/2002, E.3.), the parties to the proceedings matter, and not the parties to the contract (disputed)
-> both A and B (the parties to the proceedings) have their seat in CH
Result: PILA is not applicable (view of the SFT)
3. no exclusion (Art 176 (2) PILA)?
in casu not pertinent

**LLM Answer**: *(thinking tokens omitted)*
Under Swiss law, the validity of an arbitration agreement is governed by the **PILA (Swiss Private International Law Act)**. Specifically, **Art. 193 PILA** stipulates that the validity of an arbitration agreement is determined by: 1. The law chosen by the parties for the arbitration agreement (if specified); or 2. Failing such a choice, the law of the place where the award is to be made (i.e., the seat of arbitration).
In this case, the arbitration clause provides for arbitration in **Switzerland** as an alternative dispute resolution mechanism. Since the arbitration agreement does not explicitly designate a governing law (point 1), the validity must be assessed under the law of the **seat of arbitration** (point 2).
Here, the seat is **Switzerland**, so the validity of the arbitration agreement is determined by **Swiss law**. The reference to French courts as an alternative forum does not affect this analysis, as the arbitration clause itself is part of the contract, and its validity is evaluated independently under the PILA.
**Conclusion:** The PILA (Art. 193 PILA) applies, and the validity of the arbitration agreement is governed by **Swiss law**.
**Relevant legal provisions:** - **Art. 193 PILA**: Governs the validity of arbitration agreements. - **Art. 115 PILA** (if the contract's governing law is relevant): The contract's governing law would typically apply to the validity of the arbitration agreement if chosen, but in the absence of such a choice, the seat of arbitration (Swiss law) prevails.
No further considerations are required under the given facts.

**LLM Judgment**: 1. **Explanation**: The Model's Answer contains several missteps in its analysis compared to the Reference Answer and does not align with the precise legal reasoning or structure required by Swiss law. Key issues include:
- **Relevant Statutory Framework**: While the Model mentions Art. 193 PILA (a provision governing foreign awards), it is incorrect and completely irrelevant here. The Reference Answer makes it clear that the correct provision is **Art. 176 PILA** (Chapter 12 PILA), which applies to international arbitration with its seat in Switzerland. The Model fails to identify the operative provision and instead applies a legal framework geared toward different issues, which leads to a fundamentally flawed approach.
- **Lack of Consideration of Art. 176 (1)**: The Reference Answer correctly notes that Art. 176 (1) PILA applies only if the arbitration's seat is in Switzerland and at least one party is domiciled outside Switzerland. The Model fails to analyze whether the seat of arbitration has been validly determined in the agreement or

whether Chapter 12 PILA is rendered inapplicable due to the domicile of all parties to the dispute (A and B) being in Switzerland.
- **Overemphasis on Governing Law of the Contract**: The Model's discussion dwells unnecessarily on the general governing law of the contract and the invocation of Art. 115 PILA. This is irrelevant because the governing law of the entire contract does not automatically determine the validity of the arbitration agreement, and Art. 115 PILA is not cited in the Reference Answer as being directly relevant here.
- **Failure to Discuss Interpretation of the Arbitration Clause**: The Reference Answer raises critical questions about the **interpretation of the dispute resolution mechanism**, particularly the ambiguous "arbitration in Switzerland or French courts" language. The Model ignores these complexities and does not consider the potential invalidity of the arbitration agreement due to an uncertain seat of arbitration.
- **Not Addressing SFT Jurisprudence**: The Reference Answer references the Swiss Federal Tribunal (SFT) case law regarding when Chapter 12 PILA applies, noting that it depends on the parties to the proceedings rather than the contract. This is entirely omitted in the Model's Answer.
2. **Constructive Feedback**: To improve, the following steps would be essential: - Pay attention to the specific provisions cited in the question and accurately analyze the issue based on the correct section of the PILA (in this case, Art. 176 PILA rather than Art. 193). - Structure the response systematically: (1) the seat of arbitration, (2) the domicile of the parties to the proceedings at the time of the arbitration, (3) any exclusion via Art. 176 (2) PILA. Follow the structure laid out in the Reference Answer to ensure no legal point is overlooked. - Avoid irrelevant discussions, such as those regarding Art. 115 PILA or excessive speculation about the general governing law for the contract, unless clearly connected to the arbitration clause's validity. - Incorporate jurisprudential perspectives, such as those from the Swiss Federal Tribunal, where indicated in the Reference Answer. This strengthens the legal analysis and ensures practical alignment with current Swiss law. - Address the ambiguity in the arbitration clause's wording (e.g., whether the absence of a specific arbitration seat might render the arbitration agreement invalid).
3. **Correctness Score**: The Model fails to correctly identify the applicable provisions, misses key issues raised by the Reference Answer, and provides an incomplete and legally flawed analysis. It does not sufficiently fulfill the requirements of accuracy, completeness, and coherence in legal reasoning.
Correctness score: **[[0.2]]**.

### E.4    PROMPT TO ANSWER MCQS

We use the following prompt to answer MCQs in our experiments:

You are an expert in {course_name} and address legal issues in a structured, exam-style manner. You are given a multiple-choice question, where only one choice (e.g., A, B, C, etc.) is correct. Assume Swiss law applies unless specifically stated otherwise. If the context of the course justifies it, consider legal frameworks beyond Swiss law as well.
Please reason through the question step by step, using a chain-of-thought approach:
- Clarify the facts: Briefly restate or highlight the key facts in the question to anchor your reasoning.
- Issue Identification: What legal issue(s) arise from the facts?
- Rule Explanation: What legal rules or principles are relevant, and what are their sources (e.g., statutes, case law, doctrine)?
- Application and Reasoning: Apply the relevant rules to the facts, carefully weighing any ambiguities, exceptions, or competing interpretations. - Eliminate Incorrect Answers: Briefly explain why each incorrect answer is wrong or less convincing.
- Conclusion: Clearly state the correct answer choice (e.g., A, B, C, etc.[a]) with a brief justification for why it best fits the legal analysis.
Format your final answer as follows:
Answer: ###C###
Question: {question}
Answer:

---
[a]For the robustness check involving more than 26 answer choices, we instead numbered the answer choices.

**Reference Answer**: A

## F  INFERENCE HYPERPARAMETERS, COSTS, AND LLM ENDPOINT

**Inference Hyperparameter.**  For DeepSeek-R1 and QwQ-32B we set temperature to 0.6. For O3-mini, we set reasoning effort to high. For Claude-3.7-Sonnet, we set reasoning budget to 4096. For all reasoning LLMs, we set the max generation length to 8192, include reasoning and output tokens.

**LLM Endpoints.**  For all small conventional LLMs (7 to 14B), we use vLLM (Kwon et al., 2023) and one A100 GPU to do local inference. For close-sourced LLMs, we use the corresponding Official APIs. For the rest of LLMs, we use the Together AI API.

**Computational Cost.**  All small LLMs take roughly 7 A100 GPU hours to run full LEXam. For API costs, we spend 100 USD using GPT-4o to judge all LLMs' performance. For inference, we spend approximately 80, 20, 150 and 70 USD on OpenAI, DeepSeek, TogetherAI, and Anthropic LLMs, repectively. For Gemini-2.5-Pro, we rely on the free bonus for new users.

## G  ANNOTATION GUIDELINES, JUDGE MODEL EVALUATIONS, AND FULL ALT-TEST RESULTS

### G.1  EXPERT RECRUITMENT AND EVALUATION DATA SELECTION

We recruited three Swiss legal experts with PhD-level training, either in progress or completed, each of whom independently evaluated 50 examples, requiring approximately 10 hours of work per person. This substantial commitment reflects the open-ended nature of the questions, which demand both deep understanding and careful analysis of free-form texts. We selected this sample size because it aligns with the recommended leave-one-out design for the Alt-test and is a common standard in studies requiring intensive expert annotation.

The 50 questions were randomly sampled from the full pool of available questions. Although the selection was random, we ensured broad representativeness. The sample includes both German (n = 37) and English (n = 13) questions and reflects the main legal jurisdictions in similar proportions: Swiss law (n = 35), international law (n = 9), and generic law (n = 6). In addition, the sample covers 32 subdomains spanning private, public, criminal, and interdisciplinary legal areas. The distribution therefore mirrors the overall structure of the dataset.

### G.2  ANNOTATION PROTOCOL

All annotations were carried out by three legally trained coauthors, under the supervision of a fourth coauthor who is a law professor. Each expert annotated the same 50 question–answer pairs. The annotation task was perceived as largely self-explanatory, and no extensive calibration sessions were necessary. One of the original 50 items was replaced during the process due to concerns about it not being fully self-contained.

To ensure consistency, annotators were provided with the exact inference prompt used to generate the LLM answers as well as the evaluation prompt shown to the LLM judge. Additionally, all key instructions were summarized on a one-page slide that outlined the evaluation principles:

> - **Context:** The questions span a broad range of Swiss legal subjects. The LLM answers come from models of varying quality, including intentionally weak baselines.
> - **Objective:** Assign a score from 0 to 10 based on the match between the LLM-generated answer and a model solution.
> - **Scoring Guidelines:**
>   - **10 points:** Fully matches the model solution.
>   - **0 points:** Fails to match any relevant part.
>   - **Intermediate scores:** For example, 5 points if approx. 50% of relevant content is reproduced, 7 points for about 70% match, and so on.
> - **Handling Deviations:**

> – Additional content not in the solution is *not penalized* if legally sound.
> – Incorrect or misleading content is penalized proportionally to its severity.

The annotators noticed that unlike human law students who normally have clear concepts on the legal regulations they are referencing, some (probably weaker) LLMs produce meaningless strings next to correct legal reasoning. A particularly common issue were *hallucinated references* to laws or court decisions that sounded plausible but did not exist. A straightforward extension to LEXAM is to include an evaluation on reference selection and retrieval.

## G.3    FULL EXPERT-AGREEMENT BREAKDOWN

Table 11 shows the full pairwise agreement metrics among the three legal experts. We report Pearson correlation, quadratic weighted Cohen's $\kappa$, and mean absolute error (MAE). The aggregate values are also reported in the main paper.

Table 11: Inter-annotator agreement among the three legal experts.

|  | Pearson $r$ | Quadratic weighted $\kappa$ | MAE |
|---|---|---|---|
| Pair 1 | 0.71 | 0.58 | 1.74 |
| Pair 2 | 0.66 | 0.45 | 1.96 |
| Pair 3 | 0.75 | 0.45 | 2.14 |
| **Aggregate** | **0.70** | **0.49** | **1.95** |

Annotator pairs show consistently strong correlation ($r = 0.66$–$0.75$), with moderate-to-substantial agreement in terms of $\kappa$. Overall, the data supports the reliability of human scores.

## G.4    EXPERIMENT DESIGN ON LLM JUDGE AND DETAILED RESULTS ON ALT-TEST

Previous research on LLM-as-a-Judge has identified a major drawback: judge models may exhibit self-bias or family-bias, systematically providing overly favorable evaluations of their own outputs or of outputs from models within the same family (Panickssery et al., 2024; Spiliopoulou et al., 2025). Inspired by these studies, we argue that an ideal judge model should fulfill three desiderata. First, it should achieve performance comparable to that of human experts, verified through well-established statistical methods. Second, it should be impartial, particularly when evaluating its own outputs or those from the same model family. Third, its inference should be reasonably cost-efficient, for example by using open-source models that can run at least partially locally.

We present our experiment results using twelve settings, using both closed and open models of various sizes and evaluate the results with 50 examples labeled by domain experts and summarize the full results in Table 12. We compare LLM judges at two tolerance levels. Among stand-alone models, Claude-4-Sonnet and Gemini-2.5-Pro achieve the best performance, each yielding a perfect winning rate of 1.00 in the Alt-test. In addition DeepSeek-R1 and GPT-4o pass the Alt-test with a winning rate of 0.67, indicating that they can serve as a reliable judge within our framework. However, a stand-alone judge may still exhibit self-bias or family bias, and deploying state-of-the-art commercial models entails high costs. As an alternative, we experiment with different ensemble strategies using assessments from GPT-4o, DeepSeek-V3, GPT-OSS-120B, Qwen3-32B, and Phi-4—models with relatively low costs. In particular, we find that the most effective ensembling strategy is to pool the minimum score across models, which significantly outperforms each model's individual judgment quality, achieving a winning rate of 1.00 and an advantage probability of up to 0.76, as shown in Table 12. Due to cost considerations, our main results in Section 4 rely on an ensemble of GPT-4o, DeepSeek-V3, and Qwen3-32B,the most cost-efficient combination that achieves a winning rate of 1.0 at a tolerance level of 0. A comparison between the ensemble scores and the individual assessments of the three models is provided in Table 14.

Table 12: Winning rate $\omega$ and advantage probability $\rho$ for various LLM judges at two $\varepsilon$ levels. Bold $\rho$ and $\omega$ marks the best-performing judge models; underlined $\omega$ indicates $\omega \geq 0.5$, above the threshold to pass the test.

| Model | $\varepsilon$ | Winning Rate $\omega$ | Advantage Probability $\rho$ |
|---|---|---|---|
| Claude-4-Sonnet | 0.00 | **1.00** | **0.780** |
| | 0.15 | **1.00** | **0.780** |
| *Ensemble* | 0.00 | **1.00** | 0.760 |
| *{GPT-4o, Qwen3-32B, GPT-OSS-120B}* | 0.15 | **1.00** | 0.760 |
| Gemini-2.5-Pro | 0.00 | **1.00** | 0.760 |
| | 0.15 | **1.00** | 0.760 |
| *Ensemble* | 0.00 | **1.00** | 0.740 |
| *{GPT-4o, DeepSeek-V3, GPT-OSS-120B, Phi-4}* | 0.15 | **1.00** | 0.740 |
| *Ensemble* | 0.00 | **1.00** | 0.713 |
| *{GPT-4o, DeepSeek-V3, Qwen3-32B}* | 0.15 | **1.00** | 0.713 |
| *Ensemble* | 0.00 | 0.67 | 0.700 |
| *{GPT-4o, DeepSeek-V3}* | 0.15 | **1.00** | 0.700 |
| DeepSeek-R1 | 0.00 | 0.67 | 0.700 |
| | 0.15 | **1.00** | 0.700 |
| *Ensemble* | 0.00 | 0.67 | 0.693 |
| *{GPT-4o, Qwen3-32B}* | 0.15 | **1.00** | 0.693 |
| GPT-4o | 0.00 | 0.67 | 0.660 |
| | 0.15 | **1.00** | 0.660 |
| *Ensemble* | 0.00 | 0.00 | 0.607 |
| *{DeepSeek-V3, Qwen3-32B}* | 0.15 | **1.00** | 0.607 |
| DeepSeek-V3 | 0.00 | 0.00 | 0.573 |
| | 0.15 | 0.33 | 0.573 |
| Qwen3-32B | 0.00 | 0.00 | 0.527 |
| | 0.15 | 0.33 | 0.527 |

### G.5 ALTERNATIVE ANNOTATOR TEST: NAIVE BASELINES

To contextualize LLM performance, we applied the alt-test to naive baselines that assign the same constant score (from 1 to 10) across all 50 items. Unlike normally distributed random baselines, these highlight how well a system that "guesses" a fixed level of quality would fare.

Table 13 shows that constant-score baselines never achieve $\omega \geq 0.5$ at either $\varepsilon$ level. Their advantage probabilities $\rho$ peak around 0.507 for mid-range guesses ("Always 3" and "Always 4") but remain well below the LLM judges.

### G.6 FULL RESULTS FOR JUDGE MODEL ENSEMBLING

We present in Table 14 the full results on model performance for long-form open questions, along with bootstrapped standard errors. Qwen3-32B demonstrates self-bias, while both Qwen3-32B and GPT-4o further exhibit family bias.

## H QUANTITATIVE ANALYSES ON CROSS-LINGUISTIC CONSISTENCY OF JUDGE OUTPUTS

We further conducted two analyses to quantitatively analyze the cross-lingual consistency of judge outputs.

First, we measured the Pearson correlation between four selected judge models' scores (GPT-4o, Gemini-2.5-Pro, DeepSeek-R1, and Claude-4-Sonnet) and the average human expert scores separately

Table 13: Alternative Annotator Test results for naive constant-score baselines.

| Baseline | $\varepsilon$ | Winning Rate $\omega$ | Advantage Probability $\rho$ |
|---|---|---|---|
| Always 0 | 0.00 | 0.00 | 0.353 |
| Always 0 | 0.15 | 0.00 | 0.353 |
| Always 1 | 0.00 | 0.00 | 0.413 |
| Always 1 | 0.15 | 0.00 | 0.413 |
| Always 2 | 0.00 | 0.00 | 0.407 |
| Always 2 | 0.15 | 0.00 | 0.407 |
| Always 3 | 0.00 | 0.00 | 0.507 |
| Always 3 | 0.15 | 0.00 | 0.507 |
| Always 4 | 0.00 | 0.00 | 0.507 |
| Always 4 | 0.15 | 0.00 | 0.507 |
| Always 5 | 0.00 | 0.00 | 0.480 |
| Always 5 | 0.15 | 0.00 | 0.480 |
| Always 6 | 0.00 | 0.00 | 0.447 |
| Always 6 | 0.15 | 0.00 | 0.447 |
| Always 7 | 0.00 | 0.00 | 0.393 |
| Always 7 | 0.15 | 0.00 | 0.393 |
| Always 8 | 0.00 | 0.00 | 0.280 |
| Always 8 | 0.15 | 0.00 | 0.280 |
| Always 9 | 0.00 | 0.00 | 0.207 |
| Always 9 | 0.15 | 0.00 | 0.207 |
| Always 10 | 0.00 | 0.00 | 0.133 |
| Always 10 | 0.15 | 0.00 | 0.133 |

for English and German questions. The experiments are conducted using the 50 examples annotated by three legal experts, the sample for the Alt-test. As summarized in Table 15, the results indicate very strong correlations for English, and slightly lower, but still strong, correlations for German. The findings are consistent across all four models. These results suggest that the models' judgments are closely aligned with those of human experts in both languages.

Second, we performed a univariate linear regression to assess whether language (English vs. German) significantly explains differences between models' and human scores. The analysis yielded non-significant effects of language, with language accounting for only a small portion of the variance, as shown in Table 16. Moreover, Gemini-2.5-Pro exhibits the lowest $R^2$, indicating highest cross-linguistic consistency. This suggests that any differences in models' performance across languages are minimal and not statistically significant in our data. Overall, these quantitative results support the cross-linguistic robustness of the two models' judging performance.

## I QUALITATIVE ANALYSIS OF COMMON FAILURE MODES ACROSS DIFFERENT LEGAL DOMAINS

We further conducted qualitative analysis of common failure modes across different legal domains to provide insights for improving legal AI systems. Particularly, we examined models' evaluations and found that the most frequent failure modes were (a) flawed legal reasoning; (b) references to incorrect or even non-existent laws or court decisions; (c) weak German/multilingual proficiency, (d) frequent misinterpretation of the legal question; (e) misrepresentation of statutory content; and (f) overconfident but shallow doctrinal reasoning.

Failure modes (a) and (b) are well-documented challenges for LLMs in legal domains and appear variably across models, rather than being tied to specific questions or topics. On the other hand, failure mode (c) occurred mostly in smaller models, which often mixed languages and produced incoherent outputs. We further observed that many errors stem from the models' failure to address the correct question. For instance, when asked *"Is the PILA applicable to determine validity?"*, models instead responded by stating which law governs the validity of the arbitration agreement, a related but ultimately irrelevant issue. A further common error arises when models paraphrase statutory provisions inaccurately, leading to misinterpretations of the underlying legal content. Finally, we frequently observe overconfident yet superficial doctrinal reasoning, where models offer authoritative-

Table 14: Performance of models on long-form open questions with bootstrapped standard errors. Bold numbers indicate the best scores within each group, and underlined numbers indicate the second-best scores. Wavy underlined numbers mark the best score in a row that is assigned either by the same LLM or by a model from the same family, suggesting self-bias or family bias. The ensemble model integrates GPT-4o, DeepSeek-V3, and Qwen3-32B.

| | Model | Open Questions — Judge Score (± S.E) | | | |
| | | Ensemble | GPT-4o | Deepseek-V3 | Qwen3-32B |
|---|---|---|---|---|---|
| Reasoning | GPT-5 | **70.20 (± 0.41)** | 78.09 (± 0.35) | **76.47 (± 0.40)** | 77.60 (± 0.37) |
| | Gemini-2.5-Pro | 67.40 (± 0.51) | **82.23 (± 0.35)** | 73.95 (± 0.51) | 75.74 (± 0.38) |
| | Claude-3.7-Sonnet | 62.86 (± 0.51) | 77.63 (± 0.37) | 69.32 (± 0.52) | 71.40 (± 0.38) |
| | Claude-4.5-Sonnet | 62.76 (± 0.43) | 70.96 (± 0.42) | 67.82 (± 0.42) | 71.33 (± 0.41) |
| | GPT-5-mini | 60.32 (± 0.45) | 70.61 (± 0.42) | 63.81 (± 0.45) | 72.72 (± 0.41) |
| | DeepSeek-V3.2-Exp | 57.42 (± 0.45) | 64.97 (± 0.45) | 63.68 (± 0.45) | 67.08 (± 0.43) |
| | DeepSeek-V3.2-reasoner | 56.53 (± 0.45) | 67.36 (± 0.42) | 60.64 (± 0.47) | 69.70 (± 0.42) |
| | DeepSeek-R1 | 55.91 (± 0.51) | 68.40 (± 0.49) | 63.83 (± 0.50) | 65.95 (± 0.43) |
| | Gemini-3-Pro-preview | 55.38 (± 0.64) | 63.47 (± 0.67) | 58.80 (± 0.68) | 63.94 (± 0.69) |
| | GPT-OSS-120B | 51.74 (± 0.46) | 60.86 (± 0.48) | 56.44 (± 0.46) | 65.47 (± 0.43) |
| | O3-mini | 48.13 (± 0.49) | 55.33 (± 0.44) | 58.72 (± 0.53) | 61.87 (± 0.40) |
| | Qwen3-235B | 47.25 (± 0.46) | 53.40 (± 0.47) | 54.24 (± 0.50) | 60.11 (± 0.42) |
| | QwQ-32B | 44.36 (± 0.53) | 52.86 (± 0.57) | 53.27 (± 0.52) | 58.20 (± 0.45) |
| | Qwen3-Next | 43.37 (± 0.48) | 48.78 (± 0.51) | 50.17 (± 0.48) | 55.12 (± 0.47) |
| | Qwen3-32B | 40.00 (± 0.43) | 45.56 (± 0.46) | 44.85 (± 0.43) | 56.71 (± 0.43) |
| | GPT-OSS-20B | 32.12 (± 0.37) | 35.40 (± 0.41) | 39.21 (± 0.38) | 51.69 (± 0.42) |
| | GPT-5-nano | 27.25 (± 0.63) | 31.07 (± 0.69) | 29.61 (± 0.67) | 34.04 (± 0.74) |
| Large | GPT-4.1 | **57.50 (± 0.51)** | 68.18 (± 0.40) | **67.08 (± 0.53)** | 67.50 (± 0.42) |
| | GPT-4o | 56.93 (± 0.48) | 66.24 (± 0.39) | 66.09 (± 0.51) | 67.41 (± 0.38) |
| | DeepSeek-V3.2-chat | 55.99 (± 0.45) | 67.43 (± 0.43) | 60.40 (± 0.46) | 69.65 (± 0.42) |
| | DeepSeek-V3 | 52.53 (± 0.48) | 59.97 (± 0.43) | 62.54 (± 0.51) | 64.61 (± 0.40) |
| | Llama-4-Maverick | 47.25 (± 0.46) | 55.35 (± 0.39) | 57.43 (± 0.51) | 59.56 (± 0.38) |
| | Llama-3.1-405B-it | 43.14 (± 0.41) | 48.91 (± 0.39) | 53.59 (± 0.46) | 55.38 (± 0.38) |
| | Llama-3.3-70B-it | 41.27 (± 0.41) | 46.94 (± 0.37) | 51.45 (± 0.47) | 54.29 (± 0.37) |
| | Apertus-70B | 34.70 (± 0.39) | 44.27 (± 0.41) | 37.23 (± 0.42) | 50.64 (± 0.37) |
| Small | GPT-4.1-mini | **54.58 (± 0.43)** | 62.72 (± 0.42) | **60.74 (± 0.46)** | 64.47 (± 0.40) |
| | GPT-4.1-nano | 43.68 (± 0.41) | 49.46 (± 0.43) | 49.27 (± 0.44) | 56.74 (± 0.39) |
| | GPT-4o-mini | 42.55 (± 0.39) | 48.58 (± 0.38) | 52.91 (± 0.45) | 55.04 (± 0.37) |
| | Gemma-3-12B-it | 41.29 (± 0.48) | 50.89 (± 0.51) | 48.97 (± 0.49) | 55.22 (± 0.42) |
| | Phi-4 | 38.54 (± 0.42) | 43.45 (± 0.40) | 49.80 (± 0.47) | 54.03 (± 0.39) |
| | Gemma-2-9B-it | 27.41 (± 0.37) | 30.75 (± 0.37) | 38.96 (± 0.44) | 43.23 (± 0.38) |
| | EuroLLM-9B-it | 22.95 (± 0.35) | 25.68 (± 0.35) | 32.30 (± 0.41) | 38.17 (± 0.39) |
| | Apertus-8B | 22.44 (± 0.41) | 27.20 (± 0.44) | 24.74 (± 0.45) | 34.88 (± 0.52) |
| | Qwen2.5-7B-it | 16.67 (± 0.29) | 18.72 (± 0.29) | 26.57 (± 0.37) | 33.16 (± 0.36) |
| | Ministral-8B-it | 14.88 (± 0.32) | 16.93 (± 0.34) | 24.77 (± 0.41) | 27.25 (± 0.42) |
| | Llama-3.1-8B-it | 10.00 (± 0.26) | 11.40 (± 0.27) | 17.62 (± 0.34) | 24.79 (± 0.34) |

sounding conclusions (e.g., *"French courts do not affect the analysis"*) that are logically weak and fail to engage with the doctrinal and interpretive nuances of the case.

In addition, we observed that smaller models demonstrate weaker proficiency in German and multilingual tasks compared to larger models. Some produce incoherent outputs or mix languages, resulting in nonsensical responses, an issue unique to smaller models.

We also found that GPT-4o was the most lenient judge (+1.19 points vs. human average), while Claude-4-Sonnet and Gemini-2.5-Pro were stricter (-0.39 and -0.33, respectively), and DeepSeek-R1 was moderately generous (+0.69). Higher discrepancies between model and human scores tended to occur on items where the human annotators themselves disagreed more.

Table 15: Cross-linguistic comparison of judge models

| Model | Pearson R (EN) | P Value (EN) | Pearson R (DE) | P Value (DE) |
|---|---|---|---|---|
| GPT-4o | 0.923 | 7.01e-06 | 0.761 | 4.67e-08 |
| Gemini-2.5-Pro | 0.908 | 1.79e-05 | 0.786 | 8.07e-09 |
| DeepSeek-R1 | 0.937 | 2.30e-06 | 0.722 | 4.64e-07 |
| Claude-4-Sonnet | 0.877 | 8.18e-05 | 0.840 | 7.89e-11 |

Table 16: Statistical tests of the language effect on the score differences

| Model | P Value | $R^2$ |
|---|---|---|
| GPT-4o | 0.136 | 0.046 |
| Gemini-2.5-Pro | 0.448 | 0.012 |
| DeepSeek-R1 | 0.303 | 0.022 |
| Claude-4-Sonnet | 0.066 | 0.069 |

## J  HUMAN PERFORMANCE ON LEXAM

**Open Questions.** After conducting a human performance experiment on a subset of open-ended questions, we conclude that a direct comparison between human and model performance on these questions is not appropriate, for two reasons. First, human-written answers differ substantially from LLM-generated responses, resulting in a distributional shift. Law students typically operate under strict time constraints and are trained to provide concise answers, whereas model-generated responses are not subject to such limitations. In practice, human experts also find that scores assigned by judge models are less well aligned with their own evaluations when assessing human answers, because the judging prompts are primarily designed to evaluate LLM outputs. Designing a separate prompt specifically for human answers would introduce a different evaluation standard, thereby undermining the validity of a direct comparison. Second, due to the structure of legal education, students are not expected to take courses across all legal domains. A representative sample covering the full breadth of topics in the dataset would therefore impose an unduly broad and unrealistic burden on any individual law student.

**MCQ-4.** To complement the findings, we conducted an additional human performance experiment on a subset of four-option multiple-choice questions in Swiss law (30 questions in total). The subset was answered by a Swiss Bachelor-level law student and a doctoral-level jurist. The Bachelor-level student achieved 60% accuracy, while the doctoral-level lawyer achieved 80%.

Although the sample size remains limited, these results provide a preliminary indication that domain-specific legal training substantially improves performance. In particular, advanced legal expertise appears to outperform current reasoning models on this subset of Swiss law MCQs.

## K  FUTURE PLANS FOR DATA COLLECTION

The LEXAM benchmark is an important step toward evaluating long-context, process-based legal reasoning. We are preparing a second version with broader data to extend its scope, motivated by the following aspects.

LEXAM includes legal questions from a wide range of subjects in both English and German. Extending the multilingual scope remains desirable, particularly by incorporating questions from jurisdictions across both common law and civil law traditions. Such expansion would enable deeper investigation into cross-system and cross-cultural differences in LLM legal reasoning, which we leave for future work.

Our data curation relies on existing courses and examination materials. Given typical timelines, legal interpretations and even applicable laws may have changed since the original questions were formulated, potentially rendering some answers outdated. Our fine-grained metadata allows easy

identification of subsets affected by subsequent legal changes. Developing a mechanism to detect and update such items — similar to LiveLongBench (Wu et al., 2025) — is a promising direction.

LEXAM is intended to serve not only researchers of Swiss law but also those with broader interests in legal reasoning. While this release emphasizes Swiss law, it also incorporates data from jurisdictions such as the United States, China, and the European Union. Because law varies substantially across jurisdictions, unlike universal domains such as mathematics, the creation of high-quality, multi-jurisdictional datasets is resource-intensive. To safeguard quality, we prioritized depth over breadth in the current release.

In collaboration with leading research institutions, we are expanding the benchmark to include materials from the United States, Germany, Mainland China, Hong Kong, Israel, and additional Swiss regions, with more jurisdictions planned. An open call for contributions will follow, and new data are expected in the coming months.

