# OpenReview forum: "LEXam: Benchmarking Legal Reasoning on 340 Law Exams"
_ICLR.cc/2026/Conference — ICLR 2026 Poster_

### Official Review · Reviewer_zGey · 2025-10-20

**Soundness:** 3
**Presentation:** 4
**Contribution:** 3
**Rating:** 6
**Confidence:** 4

**Summary:**

The paper introduces LEXAM, a benchmark evaluating the legal reasoning capabilities of large language models (LLMs). The dataset comprises 4,886 law exam questions sourced from 340 law exams across 116 courses at the University of Zurich, covering a range of legal subjects in both English and German. The authors propose an ensemble LLM-as-a-Judge evaluation paradigm with expert-crafted prompts to evaluate the reasoning quality of legal reasoning and validated it against human expert assessments.

**Strengths:**

1. The idea of evaluation legal reasoning is important.
2. The authors have conducted rigorous validation of LLM-as-judges’ outputs against expert assessments.
3. The inclusion of both English and German questions

**Weaknesses:**

1. Some details about the data collection are missing. For example, how did we select the 385 questions for the auxiliary MCQ dataset? Randomly sampled from the 1660 MCQs? How did we get the distribution of questions? Did you apply any filtering process to the question selections or simply use all the questions from the courses? Then how did we select the courses?
2. The paper acknowledges the inherent complexity of legal reasoning, but, according the prompt shown in E.2, it seems that no special guideline based on legal expertise is incorporated in the LLM-as-judge system. If so, what’s the differences between judging general reasoning and judging legal reasoning? The experiments show that this paradigm can outperform human annotations on the 50 randomly selected questions. This means that either the question samples could be biased (e.g., overly simple) or legal experts don’t have a common agreements on the judgment of legal reasoning itself and thus a well-crafted prompt for general reasoning evaluation could outperform legal expert annotations directly.

**Questions:**

See in the weak points.

---

> ### Author Response · Authors · 2025-11-16
> **Response to Reviewer zGey**
>
> Dear Reviewer zGey,
>
> We thank you for your thoughtful feedback and for highlighting
> - the importance of evaluating legal reasoning,
> - the rigor of our LLM-as-a-Judge validation, and
> - the multilingual scope of LEXam.
>
> We address the raised concerns and questions below.
>
> ---
> Q1.
>
> *“How did we select the 385 questions for the auxiliary MCQ dataset?”*
>
> The 385 questions are the subset of the 1,660 MCQs that contain exactly five statements, which allows us to generate up to 32 (2⁵) answer candidates per question (as shown in the example).
>
> *“Randomly sampled from the 1660 MCQs?”*
>
> This subset is therefore not a random sample; it is determined solely by whether an MCQ provides five statements. Two authors manually inspected all selected MCQs to ensure quality, and we did not identify any low-quality items, which is consistent with the high standard of the original law-school exam questions from which they were drawn.
>
> *“How did we get the distribution of questions?”*
>
> The 385 questions come from ten courses, and we will include a distribution table across these ten courses in the Appendix.
>
> *“Did you apply any filtering process to the question selections or simply use all the questions from the courses?”*
>
> We keep all open questions and MCQs from the original exam papers, which correspond to courses.
>
> *“Then how did we select the courses?”*
>
> The courses were included because they fall under one of four higher-level categories: private law, public law, criminal law, and interdisciplinary. Two legal experts then merged courses that are conceptually very close, yielding 78 legal subdomains, as detailed in Section 2.1 (lines 135–143).
>
> ---
> Q2.
>
> *“No special guideline based on legal expertise is incorporated in the LLM-as-judge system. If so, what’s the difference between judging general reasoning and judging legal reasoning?”*
>
> We respectfully argue that **legal expertise is extensively incorporated in our prompt** for two reasons:
> - Firstly, we provide **professor-written** gold-standard solutions that are, by nature of law-school exams, highly precise in their legal reasoning structure (issue spotting, rule selection, application, treatment of alternatives).
> - Secondly, as described in Sec. 3.1, the prompt was **iteratively developed by two authors with doctoral-level legal training**, so legal expertise is directly incorporated. It is correct that we do not define an abstract theory of best legal reasoning in the prompt. This would require engaging with a vast jurisprudential literature and is well beyond the scope of a benchmarking paper.
>
> Our approach is therefore **bottom-up**: the judge is optimized to evaluate adherence to the doctrinal structure encoded in these legal expert model answers, rather than to an a priori philosophical definition of legal reasoning. This includes domain-specific behaviors already present in E.2, such as penalizing fabricated or incorrect statutory citations and respecting legally permissible alternative solution paths (the +/– markers). These elements distinguish our setup from general reasoning evaluation, where multiple doctrinally valid solutions, unlike in law, rarely exist.
>
> We will clarify this bottom-up design choice more explicitly in the revised version.
>
> *“[...] the question samples could be biased (e.g., overly simple) [...]”*
>
> Bias or Simplicity in Question Sample: We respectfully argue that the question sample is NOT overly simple, as they are **uniformly randomly sampled** as described in Appendix G.1. It is representative of the full dataset in both domain coverage and difficulty, and systematic simplicity or sampling bias is therefore highly unlikely.
>
> Our legal experts also confirmed that the sampled questions are **very difficult and require several years of rigorous legal training to answer**.
>
> *“[…] legal experts don’t have a common agreement on the judgment of legal reasoning itself [...]”*
>
> The legal expert annotations achieve a **substantial inter-annotator agreement (Pearson r = 0.70)**, which lies squarely within the stable range for which the Alt-Test is designed. Certain disagreement is expected in open-ended legal reasoning, especially when multiple doctrinally permissible solution paths exist. However, both the sample size and the level of annotator consistency satisfies the conditions under which the authors of the Alt-Test intend their method to be applied.
>
> *“[...] and thus a well-crafted prompt for general reasoning evaluation could outperform legal expert annotations directly.”*
>
> We also want to emphasize that our prompt is **meticulously tailored for grading LEXam questions with legal expertise**, rather than for general reasoning evaluation.
>
> ---
> We hope our answers above help clarify the points you raised and address the questions in your review. If there are any aspects that would benefit from further explanation, please feel free to let us know, and we would be happy to elaborate.

---

### Official Review · Reviewer_ASGZ · 2025-10-30

**Soundness:** 3
**Presentation:** 4
**Contribution:** 3
**Rating:** 8
**Confidence:** 5

**Summary:**

The paper presents a new dataset consisting of ~4500 examples with multi-choice and long form questions taken from law exams at the University of Zurich. The LEXam benchmark covers English and German languages and multiple jurisdictions from Switzerland, Europe and international. It also provides an extensive array of baseline experiments covering multiple open and closed weight LMs of different sizes and families. Results show that the questions are challenging (GPT-5 achieving best performance ~70% for open and ~63% for multi-choice).

**Strengths:**

- Valuable resource that will enable further research in LLM capabilities in a very critical domain such as law
- Extensive benchmarking and analysis of results.

**Weaknesses:**

- Not clear what the human performance is on the dataset. It might be a good idea to include it in Table 1.

**Questions:**

- See weaknesses.

---

> ### Author Response · Authors · 2025-11-14
> **Response to Reviewer ASGZ**
>
> Dear Reviewer ASGZ,
>
> We thank you for the positive feedback and for recognizing our dataset as a valuable resource that can support further research on LLM capabilities in the critical domain of law, as well as the extensive benchmarking and analysis we conducted.
>
> Furthermore, we appreciate your suggestions very much and would like to address the issue regarding human performance. Due to institutional and regulatory constraints, we unfortunately do not have access to individual student performance data on open questions from the University of Zurich Faculty of Law. In addition, obtaining such data directly from students is not feasible. However, we have contacted multiple professors and plan to collect aggregate performance results for selected courses in upcoming semesters. Moreover, because our multiple-choice questions are pre-processed (as detailed in Section 3.2), the current MCQ format does not appear in previous exams. We therefore plan to recruit a small number of law students to evaluate human performance on these MCQs. We will elaborate on the planed experiments in the revised paper.
>
> If any questions remain, please feel free to let us know, and we would be glad to clarify.

---

### Official Review · Reviewer_PxoQ · 2025-10-31

**Soundness:** 2
**Presentation:** 3
**Contribution:** 3
**Rating:** 6
**Confidence:** 3

**Summary:**

This paper introduces LEXAM, a comprehensive benchmark designed to evaluate the long-form legal reasoning capabilities of LLMs. LEXAM is constructed from 4,886 law exam questions collected from 340 exams across 116 law school courses, covering a wide range of subjects and degree levels in English and German. The dataset includes 2,841 open-ended questions and 2,045 multiple-choice questions. Each open-ended question is accompanied by detailed guidance outlining the expected reasoning process, such as issue spotting, rule recall, and rule application.
Through extensive evaluation, the authors find that current LLMs perform poorly on tasks requiring structured reasoning, highlighting persistent challenges in legal understanding. The study further demonstrates the discriminative power of LEXAM in distinguishing models with different reasoning strengths. By employing an ensemble LLM-as-a-Judge framework validated by human experts, the authors provide a scalable and consistent evaluation methodology that aligns closely with expert assessments and goes beyond traditional accuracy-based metrics, offering a new paradigm for measuring legal reasoning quality.

**Strengths:**

(1) Comprehensive Real-World Dataset. The paper presents a comprehensive real-world dataset for legal reasoning, derived from 340 actual law exams across 116 courses at the University of Zurich. The dataset contains 4,886 questions in both English and German. Notably, it includes long-form, human-written reference answers accompanied by explicit guidance on the expected reasoning process—resources that remain rare in existing legal NLP benchmarks.
(2) Broad Empirical Evaluation. The authors conduct an extensive empirical study covering 26 diverse large language models, including reasoning-oriented systems such as GPT-5, DeepSeek-R1, Gemini-2.5, and Claude-3.7.
(3) Evaluation Design and Reliability. For open questions, the paper adopts an LLM-as-a-Judge framework that ensembles multiple models. This ensemble approach appears more robust than traditional single-model judging methods. For multiple-choice questions, the authors further design perturbed variants by varying the number of answer options, in order to test model robustness and sensitivity to superficial cues.

**Weaknesses:**

(1) Limited Scope and Jurisdictional Coverage. The dataset is almost entirely derived from Swiss and European law exams at the University of Zurich, using only English and German. Although the authors briefly mention cross-system differences, the dataset design remains heavily biased toward the civil-law tradition. It lacks representation of common-law jurisdictions such as the United States and civil-law contexts like China’s legal system. As a result, while the contribution is valuable, its applicability and generalizability across broader legal systems are clearly limited.
(2) Question Equivalence and Multilingual Validity. LEXAM claims to be a “multilingual legal reasoning benchmark,” but it does not clarify whether the English and German questions are parallel translations or independent items. Nor does it specify whether bilingual legal experts verified semantic and legal-effect equivalence between the two languages. The authors should explicitly state this point, as it directly affects the validity of cross-language comparisons.
(3) Evaluation Framework and Conceptual Limitations. While the proposed ensemble-based LLM-as-a-Judge approach is more robust than using a single LLM judge, it does not introduce a fundamentally new or thought-provoking evaluation paradigm. Moreover, the logic of using LLMs to generate answers, then using other (or similar) LLMs to grade those answers, and finally using those grades to infer the reasoning capability of the same class of models, raises methodological concerns about circularity and self-referential validation.
(4) Entanglement Between Legal Reasoning and Multilingual Processing. The models exhibit strong language sensitivity. As shown in Figure 5, performance in English is consistently and significantly higher than in German. This suggests that the benchmark’s evaluation signal mixes legal reasoning ability with multilingual processing and translation competence. Since the benchmark does not attempt to disentangle these two components, it becomes difficult to interpret the reported results as a clean measure of “legal reasoning” alone.

**Questions:**

(1) Are the English and German questions parallel translations, or are they completely independent question sets? If they are translations, were they verified by qualified legal professionals to ensure semantic and legal-effect equivalence?
(2) The results show a substantial English–German performance gap. How do the authors disentangle legal reasoning ability from multilingual or translation-related effects in this evaluation?
(3) How do the authors attempt to address or analyze the potential circularity of using LLMs to evaluate other LLMs’ answers—beyond the reported human consistency checks?

---

> ### Author Response · Authors · 2025-11-16
> **Response to Reviewer PxoQ (Part 1)**
>
> Dear Reviewer PxoQ,
>
> We thank you for the detailed and constructive review. We are especially grateful for the recognition of our key contributions:
> - its realism and depth as a real-world law exam dataset,
> - the inclusion of long-form human-written answers with reasoning guidance,
> - the breadth of our empirical evaluation across 26 LLMs, and
> - the reliability improvements introduced through our ensemble-based scoring framework.
>
> We now address the weaknesses (Ws) you suggested point by point:
>
> ---
> *W1. Lacking “representation of common-law jurisdictions such as the United States [...] [and] China’s legal system.”*
>
> The dataset covers both civil law and common law context. As detailed in lines 1028–1049, LEXam includes multiple courses with **international, comparative, and theoretically grounded content** that is highly relevant across legal systems. Courses such as Legal Theory, Legal Sociology, Comparative Private Law, International Commercial Arbitration, and International Human Rights Law are also central components of common-law curricula, so their inclusion extends the benchmark **beyond the (Swiss) civil law context**. In addition, the dataset contains exams focused on **U.S. law, Chinese law, and international law**.
>
> In addition, we provide rich metadata on jurisdictions, legal subdomains, and high-level legal areas. This enables users to select specific fields and balance the dataset distribution to better match their evaluation needs.
>
> We invested substantial effort in curating the 340 exams in LEXam and ensuring that the data collection process was ethically approved. It is extremely challenging for us to gather a broader set of legal exams from across the world, so we leave this expansion to future work, where we plan to extend the benchmark to additional universities in the U.S., China, Germany, and beyond. **A detailed plan for this future work is provided in Appendix J**, which we hope helps contextualize the current limitation of including only 340 exams.
>
> ---
> *W2. Whether the English and German questions are “parallel translations or independent items.”*
>
> The exams are pre-processed directly from original law school exams. Therefore, they are **independent items, not parallel translations**. We didn’t consider translating the questions, as legal translation is itself a complex task (e.g. [1]). We appreciate the suggestion to more explicitly discuss that no translation has been applied, and will clarify this in the revised version.
>
> [1] Niklaus et al. "Swiltra-bench: The Swiss legal translation benchmark." *arXiv:2503.01372* (2025).
>
> ---
> *W3. The proposed method “does not introduce a fundamentally new evaluation paradigm […] and the use of LLMs to generate answers, have other (or similar) LLMs grade them, [...] raises concerns about circularity and self-referential validation.”*
>
> We respectfully clarify that **the main contribution type** of this work is a benchmark dataset, and we **do not present our LLM-as-a-Judge ensemble as a stand-alone, novel evaluation paradigm**. Rather, we introduce it as a **practical and empirically validated solution** for open-ended legal evaluation, benchmarked against human annotations using the Alt-test framework (Sections 5.2, 5.3; Appendices G.4–G.6). The ensemble is specifically designed to reduce, and has effectively reduced, well-documented biases from individual judge models, particularly **self-bias** and **family-bias**.
>
> Regarding “self-referential validation”: we fully agree that LLM judges may have potential bias, this is **exactly why we spend huge effort in our evaluation design**, as detailed in lines 216–243 (Section 3.1) and 1663–1711 (Appendix G.4). Specifically, we have following designs to enhance and validate our LLM-based evaluation:
> - Prompt design guided by pilot study conducted by subject-matter legal experts (detailed in Appendix E.2).
> - Strict comparison of LLM vs. Human judging performance by Alt-test [2], with statistical power (detailed in Section 5.2 and Appendix G.4).
> - An ensemble of diverse LLM judges is used to reduce self-bias and family bias [3, 4] (which we believe is the self-referential issue you are referring to). In particular, it aggregates pointwise minimum scores, suppressing inflated evaluations for answers generated by the same model or family. This approach has proven effective, as detailed in Appendix G.6 and Table 13.
>
> We believe these designs will ensure the reliability of our LLM-based evaluation which is necessary for open question grading.
>
> [2] Calderon et al. "The alternative annotator test for llm-as-a-judge: How to statistically justify replacing human annotators with llms." *arXiv:2501.10970* (2025).
>
> [3] Panickssery et al. "Llm evaluators recognize and favor their own generations." *Advances in Neural Information Processing Systems* 37 (2024): 68772-68802.
>
> [4] Spiliopoulou et al. "Play favorites: A statistical method to measure self-bias in llm-as-a-judge." *arXiv:2508.06709* (2025).

---

> ### Author Response · Authors · 2025-11-16
> **Response to Reviewer PxoQ (Part 2)**
>
> ---
> *W4. “The models exhibit strong language sensitivity. [...] [It is] difficult to interpret the reported results as a clean measure of 'legal reasoning' alone.”*
>
> Thanks for raising this! However, we respectfully argue that **disentangling multilinguality and legal reasoning is outside the scope of this paper**, given the difficulty and complexity of the task. We agree that language differences may reflect both linguistic competence and legal reasoning variation. However, this is an inherent feature of multilingual legal tasks. Attempting to “disentangle” these effects would require high-quality legal translation, which itself is a non-trivial task.
>
> Therefore, we acknowledge it as a valuable direction for future research. As discussed in Appendix J, we plan to expand the dataset to additional jurisdictions and course offerings. This will allow us to directly address this question by
> - controlling for language across comparable courses (e.g., contract law taught in German vs. English) and
> - controlling for language across jurisdictions (e.g., French questions for the same course types in France vs. French-speaking Switzerland vs. Quebec, and English questions for the same course types in the U.S. vs. U.K. vs. Hong Kong).
>
> ---
> Here, we address your questions (Qs).
>
> ---
> *Q1. "Are the English and German questions parallel translations, or are they completely independent question sets? If they are translations, were they verified by qualified legal professionals to ensure semantic and legal-effect equivalence?"*
>
> As elaborated in *W2*, the English and German questions are completely independent question sets. Therefore, we do not claim semantic and legal-effect equivalence of the questions.
>
> ---
> *Q2. "How do the authors disentangle legal reasoning ability from multilingual or translation-related effects in this evaluation?"*
>
> As we clarified in *W4*, disentangling multilinguality from legal reasoning is outside the scope of this paper, as doing so would require high-quality legal translation—a non-trivial task in its own right. We therefore leave this to future work, as outlined in Appendix J, where broader data availability will allow us to pursue this direction.
>
> ---
> *Q3. "How do the authors attempt to address or analyze the potential circularity of using LLMs to evaluate other LLMs’ answers—beyond the reported human consistency checks?"*
>
> The concern about circularity or self-reference, as discussed in *W3*, primarily manifests as self-bias and family-bias. To analyze this issue and evaluate the efficacy of our proposed ensemble LLM judge, we conduct experiments using both the ensemble model and the three individual models. We show that the problem is effectively mitigated when three necessary conditions (NCs) are satisfied:
> - **NC1**: single models indeed exhibit self-bias and family-bias;
> - **NC2**: there exists an alternative model  ***M*** that effectively suppresses overly favorable evaluations for answers generated by the same model or model family; and
> - **NC3**: the model ***M*** yields scores that are better aligned with human scores.
>
> As summarized in lines 1782–1815 (Table 13), both GPT-4o and Qwen3-32B exhibit self-bias and family-bias when used as single judges, thereby satisfying NC1. Furthermore, as described in lines 235–243 (Section 3.1) and 1663–1711 (Appendix G.4), our ensemble judge aggregates pointwise minimum scores and suppresses overly favorable evaluations for answers generated by the same model or model family, thereby satisfying NC2. Finally, Table 11 (lines 1685–1711) shows that the ensemble model ***M*** produces scores that align more closely with human scores under the Alt-test, thereby satisfying NC3.
>
> Taken together, these results demonstrate that our ensemble judge satisfies all three necessary conditions and therefore effectively mitigates circularity in LLM-based evaluation. We will further refine this analysis in the revised paper, as explained here.
>
> ---
> We hope our responses above help clarify the points you raised and address the concerns in your review. If there are any aspects that would benefit from further explanation, we would be happy to elaborate.

---

### Official Review · Reviewer_UVpG · 2025-10-31

**Soundness:** 3
**Presentation:** 3
**Contribution:** 3
**Rating:** 4
**Confidence:** 4

**Summary:**

The paper introduces LEXAM, a large-scale benchmark designed to evaluate legal reasoning abilities of large language models (LLMs).
The dataset is built from 340 authentic law school exams from the University of Zurich, covering 4,886 questions (2,841 open-ended and 2,045 multiple-choice) across 116 courses and 78 subfields of law, in both English and German.

The authors propose a novel evaluation framework called LLM-as-a-Judge, an ensemble of models (GPT-4o, Qwen3-32B, DeepSeek-V3) designed to simulate human expert grading. Results show that GPT-5 achieves the highest overall performance (70.2/100 on open questions, 62.7% on multiple choice), while the ensemble grader attains agreement levels comparable to or exceeding human legal experts.

The study presents an ambitious, well-structured attempt to standardize evaluation of LLM reasoning in legal domains.

**Strengths:**

The benchmark is constructed from real law school exams rather than synthetic or reformulated items, ensuring high realism, linguistic diversity, and true reasoning difficulty.

The framework evaluates both open-ended and multiple-choice questions, tests robustness under option perturbations, and analyzes performance across languages, jurisdictions, and subfields.

The ensemble scoring system of multiple LLMs (taking minimum scores) is well-motivated and empirically validated against human expert grading. It represents a meaningful step toward scalable, consistent evaluation of open-ended legal reasoning.

Including both English and German questions, spanning Swiss and international law, greatly enhances the generalizability and research utility of the benchmark.

**Weaknesses:**

1. All data originate from Swiss civil law contexts; thus, generalization to common law systems (e.g., US/UK) is uncertain.

2. Although “LLM-as-a-Judge” performs well, it introduces dependence on closed models (GPT-4o, DeepSeek-V3) and may inherit their biases.

3. The paper reports scores but lacks detailed qualitative analysis or case studies illustrating where models succeed or fail in reasoning.

4. While the dataset is said to be authorized, the anonymization and ethical review process is not discussed in depth.

5. Quantitative comparison with existing benchmarks (LegalBench, LexGLUE, LawBench, COLIEE) is limited to descriptive discussion rather than aligned experimental results.

**Questions:**

none

---

> ### Author Response · Authors · 2025-11-14
> **Response to Reviewer UVpG (Part 1)**
>
> Dear Reviewer UVpG,
>
> We thank you for your detailed feedback and appreciate the recognition of our benchmark’s key contributions, including
> - its high realism, linguistic diversity, and genuine reasoning difficulty stemming from real law-school exams,
> - its comprehensive evaluation framework spanning open-ended and multiple-choice questions, perturbation robustness, and cross-language/jurisdiction/subfield analyses,
> - the well-motivated and empirically validated ensemble scoring approach that aligns closely with expert grading, and
> - the inclusion of both English and German questions across Swiss and international law, which substantially broadens the benchmark’s generalizability and research value.
>
> We address the questions and concerns raised below:
>
> ---
> *1. "All data originate from Swiss civil law contexts; thus, generalization to common law systems (e.g., US/UK) is uncertain."*
>
> The dataset is **not confined to Swiss civil law context**. As detailed in lines 1028–1049, LEXam includes multiple courses with **international, comparative, and theoretically grounded content** that is highly relevant across legal systems. Courses such as Legal Theory, Legal Sociology, Comparative Private Law, International Commercial Arbitration, and International Human Rights Law are also central components of common-law curricula, so their inclusion extends the benchmark beyond Swiss law in particular, and beyond the civil law context in general. In addition, the dataset contains exams focused on **U.S. law, Chinese law, and international law**. We agree that broader global coverage is important and are actively working on expanding LEXam to include more exams from common law jurisdictions in future iterations (LEXam-v2). The detailed plan for LEXam-v2 is reported in Appendix J.
>
> ---
> *2. "Although “LLM-as-a-Judge” performs well, it introduces dependence on closed models (GPT-4o, DeepSeek-V3) and may inherit their biases."*
>
> Respectfully, **our design directly addresses this concern.** As detailed in Section 3.1, we clarify that our ensemble judge integrates outputs from both **open** models (DeepSeek-V3, Qwen3-32B) and a **closed** model (GPT-4o), rather than relying on any single judge. The ensemble LLM-as-a-Judge is explicitly designed to reduce biases of individual models,such as self-bias and family bias, which we analyze in detail in lines 1724–1727 (Appendix G.6) and 1782–1815 (Table 13). As described in lines 235–243 (Section 3.1) and 1663–1711 (Appendix G.4), the ensemble aggregates point-wise minimum scores, suppressing overly favorable evaluations for answers generated by the same model or model family.
>
> Importantly, our approach **does not depend heavily on closed models; open models form the majority of the ensemble** (two of three). We also document fully open-model alternatives in Appendix G.4 and will report additional results on other open models, including QwQ-32B, GPT-OSS-120B, and Phi-4, as well as their strongest ensemble configurations. Moreover, users can run the Alt-test with any models of their choice.
>
> ---
> *3. "The paper reports scores but lacks detailed qualitative analysis or case studies illustrating where models succeed or fail in reasoning."*
>
> We included detailed qualitative analyses and case studies in Appendix I (lines 1772/1828–1825), referenced in the main text (lines 425–431). We also provided metadata-based analyses in Figures 4 and 5, which reveal systematic patterns. In particular, we identified three frequent failure modes:
> - **(a) flawed legal reasoning chains**,
> - **(b) references to incorrect or non-existent laws or court decisions**, and
> - **(c) weak German/multilingual proficiency**.
>
> Modes (a) and (b) are well-documented challenges for LLMs in legal domains and vary across models rather than being tied to specific questions. Mode (c) appears primarily in smaller models, which often mix languages and produce incoherent outputs. These analyses allow us to identify representative strengths and consistent failure patterns.
>
> **Additional qualitative analysis:** Moreover, we conducted additional qualitative analyses after submitting the paper, through which we identified three further error patterns:
> - **(d) frequent misinterpretation of the legal question** (e.g., when asked *“Is the PILA applicable to determine validity?”* models instead answered which law governs the validity of the arbitration agreement);
> - **(e) misrepresentation of statutory content** (e.g., inaccurate paraphrasing of provisions); and
> - **(f) overconfident but shallow doctrinal reasoning**, where models offer authoritative-sounding conclusions (e.g., *“French courts do not affect the analysis”*) that are logically weak and fail to engage with the doctrinal and interpretive issues in the case.
>
> We will elaborate on these findings in an additional paragraph in the revised version.

---

> ### Author Response · Authors · 2025-11-14
> **Response to Reviewer UVpG (Part 2)**
>
> ---
> *4. "While the dataset is said to be authorized, the anonymization and ethical review process is not discussed in depth."*
>
> Respectfully, we believe **this has been addressed in our paper** (as in lines 490–492 and 540–551). In particular, our dataset **does not contain any personal, sensitive, or identifying information about individuals**, but only publicly available law school exam questions and reference answers, released by the University of Zurich Faculty of Law for educational purposes (as stated in lines 490–492). Student-written answers are explicitly excluded. Furthermore, as mentioned in our Ethics Statement (lines 540–551), we obtained formal permission from the University of Zurich and used the data solely for research.
>
> Because the dataset includes no confidential content, no anonymization procedures were necessary. In addition, after consulting multiple legal experts, we confirmed that no further ethical review was required. Ethics approval is mandatory only for research involving human subjects, and our study involved neither human participants nor human data. We will clarify this rationale in the revised paper.
>
> ---
> *5. "Quantitative comparison with existing benchmarks (LegalBench, LexGLUE, LawBench, COLIEE) is limited to descriptive discussion rather than aligned experimental results."*
>
> We respectfully argue that aligning experimental results with the suggested benchmarks is **not informative**, as these benchmarks are **structurally and conceptually very different from LEXam**.
>
> **Detailed comparisons:** LEXam is constructed from full law-school exams and evaluates exam-style legal reasoning within an educational curriculum. In contrast, LegalBench, LexGLUE, LawBench, and COLIEE consist of task-specific datasets (e.g., classification, retrieval, entailment) with fixed label spaces and heavily pre-processed inputs. Each benchmark targets a narrow, well-defined task family: LegalBench aggregates many small doctrinal tasks, LexGLUE focuses on legal text classification across multiple subtasks, LawBench evaluates structured legal reasoning on synthetic or templated problems, and COLIEE centers on case-based retrieval and entailment. Therefore, while LEXam primarily assesses free-text, open-ended legal reasoning, the four benchmarks focus on structured task formulations, such as classification, retrieval, or entailment, rather than on producing full legal analyses or applying doctrines in free-text form. Even for multiple-choice questions in LEXam, comparability is limited. LEXam spans diverse legal domains (e.g., tort law, constitutional law), rather than narrow task types (e.g., legal entailment), so a direct mapping between its question types and standard benchmark tasks is neither natural nor meaningful.
>
> As a result, producing aligned experiments would require many arbitrary design decisions, such as mapping long-form answers to short labels, fragmenting multi-part questions, or reconciling incompatible metrics and evaluation protocols. These choices could substantially affect results without adding real insight and would risk creating an illusion of comparability.
>
> Our goal in the related-work section is therefore to position LEXam within the broader context: we offer a descriptive comparison in terms of jurisdiction, languages, task types, and evaluation protocol, and we empirically show, within LEXam, that current frontier models still struggle with realistic exam-style legal reasoning. This complements, rather than duplicates, existing benchmarks, which target different task formulations and evaluation regimes.
>
> ---
> Our responses above aim to clarify the points raised and address the concerns in your review. If there are any aspects that would benefit from further explanation, we are happy to elaborate.

---

### Author Response · Authors · 2025-11-26
**Paper Revised**

Dear Reviewers,

As noted in our earlier individual responses, we have revised the paper in accordance with the official comments. In particular, we:

- updated the draft as promised in the individual responses, and

- incorporated five additional reasoning models (Claude-4.5-Sonnet, DeepSeek-V3.2, GPT-OSS-120B, GPT-OSS-20B, and Qwen3-Next) in our evaluation and updated the corresponding analyses for all 31 models.

If you have any further questions or suggestions, we would be happy to address them. We look forward to continued discussion.

Sincerely,

Authors of Submission #17572

---

### Meta-Review · Area_Chair_1o28 · 2025-12-29

**Summary:**

The paper proposes LEXam -- a benchmark on legal reasoning, which the authors use to evaluate LLMs. The proposed on law exams (so, it consists of natural, human-written questions) and is somewhat diverse: it incudes questions on Swiss law as well as number of questions on international, Chinese, US law (though those are underrepresented). This benchmark surfaces limitations in legal reasoning of current LLMs.

Reviewers' concerns: see below.

**Reviewer Concerns:**

Concerns resolved by authors' rebuttal include: LLM as a judge evaluation quality, qualitative analysis of models' failures on the benchmark, ethical considerations, comparison and differences with the prior benchmarks in legal domain, clarifications on the multi-lingual nature and analysis of the benchmark.


Outstanding concern:

1. Human performance: as per review ASGZ, it would be helpful to report human expert level performance on this dataset.

Partially addressed concerns:

2. Scope of the benchmark with regards to Swiss law vs other countries.
The authors re-iterate in the rebuttal that their dataset also induces international, Chinese and US law. Additionally, in Figure 3 and in the appendix they report statistics of the dataset, and it looks like the majority of questions fall under Swiss law. It would be helpful to clarify this aspect in the main text of the camera-ready version that x% questions are on Swiss law, while y% are on Chinese law etc. -- ideally this information should be in the Introduction section under contribution list to set the scope of the dataset upfront, since this concern was raised multiple times.

3. It would be important to incorporate clarifications from the rebuttal to reviewer zGey in the paper, especially clarifications on the dataset filtering.

Given these questions are clarified in the camera-ready, I suggest a "conditional accept" for this paper.

**Reviewer Scores:**

Reviewer UVpG: I believe concerns of this reviewer were largely addressed so they would be quite likely to increase the score from 4 to 6.

Reviewer PxoQ: I think this reviewer would retain their score as 6. Their concerns were addressed but there were also no critical concerns identified.

Reviewer ASGZ: this was a one-liner review which for the most part I discarded in making a final decision. However, they raised a valid concern in their extremely short review regarding human expert evaluation which remains unresolved after rebuttal.

Reviewer zGey: I believe the concerns of this reviewer were addressed for the most part (specifically, clarification re data filtering processing and evaluation), so this reviewer would likely leave their score as 6 or update 6 -> 8 (less likely).

So I think the final scores would be: 6, 6, 6. Taking into account the detailed feedback, I suggest "conditional accept", given the authors address the outstanding reviewer concerns in the camera-ready version of the paper.

---

### Decision · Program_Chairs · 2026-01-26

Accept (Poster)